# PAC Confidence Sets for Deep Neural Networks via Calibrated Prediction

**Sangdon Park**
University of Pennsylvania
sangdonp@cis.upenn.edu

**Osbert Bastani**
University of Pennsylvania
obastani@seas.upenn.edu

**Nikolai Matni**
University of Pennsylvania
nmatni@seas.upenn.edu

**Insup Lee**
University of Pennsylvania
lee@cis.upenn.edu

## Abstract

We propose an algorithm combining calibrated prediction and generalization bounds from learning theory to construct confidence sets for deep neural networks with PAC guarantees—i.e., the confidence set for a given input contains the true label with high probability. We demonstrate how our approach can be used to construct PAC confidence sets on ResNet for ImageNet, a visual object tracking model, and a dynamics model for the half-cheetah reinforcement learning problem. [1]

## 1 Introduction

A key challenge facing deep neural networks is that they do not produce reliable confidence estimates, which are important for applications such as safe reinforcement learning (Berkenkamp et al., 2017), guided exploration (Malik et al., 2019), and active learning (Gal et al., 2017).

We consider the setting where the test data follows the same distribution as the training data (i.e., we do *not* consider adversarial examples designed to fool the network (Szegedy et al., 2014)); even in this setting, confidence estimates produced by deep neural networks are notoriously unreliable (Guo et al., 2017). One intuition for this shortcoming is that unlike traditional supervised learning algorithms, deep learning models typically overfit the training data (Zhang et al., 2017). As a consequence, the confidence estimates of deep neural networks are flawed even for test data from the training distribution since, by construction, they overestimate the likelihood of the training data.

A promising approach to addressing this challenge is *temperature scaling* (Platt, 1999). This approach takes as input a trained neural network $f_{\hat{\phi}}(y \mid x)$—i.e., whose parameters $\hat{\phi}$ have already been fit to a training dataset $Z_{\text{train}}$—which produces unreliable probabilities $f_{\hat{\phi}}(y \mid x)$. Then, this approach rescales these confidence estimates based on a validation dataset to improve their "calibration". More precisely, this approach fits confidence estimates of the form

$$f_{\hat{\phi},\tau}(y \mid x) \propto \exp(\tau \log f_{\hat{\phi}}(y \mid x)),$$

where $\tau \in \mathbb{R}_{>0}$ is a *temperature scaling* parameter that is fit based on the validation dataset. The goal is to choose $\tau$ to minimize *calibration error*, which roughly speaking measures the degree to which the reported error rate differs from the actual error rate.

The key insight is that in the temperature scaling approach, only a single parameter $\tau$ is fit to the validation data—thus, unlike fitting the original neural network, the temperature scaling algorithm comes with generalization guarantees based on traditional statistical learning theory.

Despite the improved generalization guarantees, these confidence estimates still do not come with theoretical guarantees. We are interested in producing *confidence sets* that satisfy statistical guarantees while being as small as possible. Given a test input $x \in \mathcal{X}$, a confidence set $C_T(x) \subseteq \mathcal{Y}$

---

[1]Our code is available at https://github.com/sangdon/PAC-confidence-set.

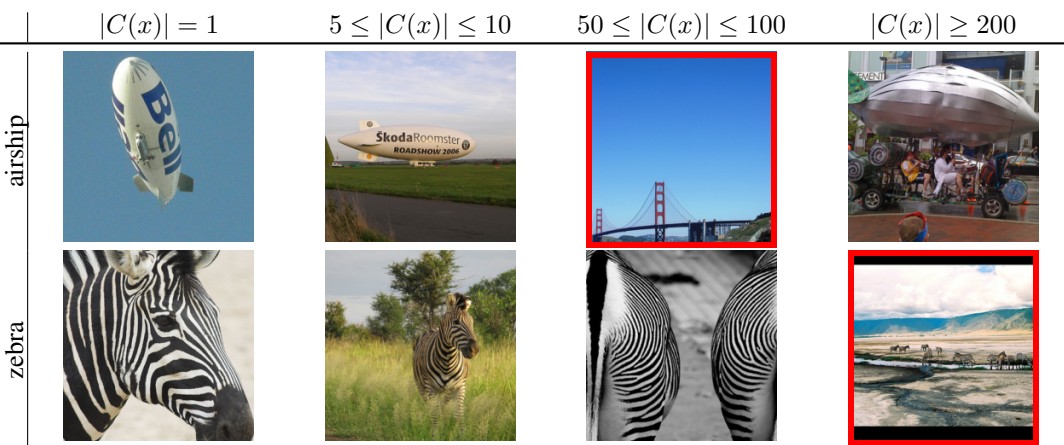

Table 1: ImageNet images with varying ResNet confidence set sizes. The confidence set sizes are on the top. The true label is on the left-hand side. Incorrectly labeled images are boxed in red.

(parameterized by $T \in \mathbb{R}$) should contain the true label $y$ for at least a $1 - \epsilon$ fraction of cases:

$$\mathbb{P}_{(x,y) \sim D}[y \in C_T(x)] \geq 1 - \epsilon.$$

Since we are fitting a parameter $T$ to based on $Z_{\text{val}}$, we additionally incur a probability of failure due to the randomness in $Z_{\text{val}}$. In other words, given $\epsilon, \delta \in \mathbb{R}_{>0}$, we aim to obtain *probably approximately correct (PAC)* confidence sets $C_T(x) \subseteq \mathcal{Y}$ satisfying the guarantee

$$\mathbb{P}_{Z_{\text{val}} \sim D^n} \left( \mathbb{P}_{(x,y) \sim D}(y \in C_T(x)) \geq 1 - \epsilon \right) \geq 1 - \delta.$$

Indeed, techniques from statistical learning theory (Vapnik, 1999) can be used to do so (Vovk, 2013).

There are a number of reasons why confidence sets can be useful. First, they can be used to inform safety critical decision making. For example, consider a doctor who uses prediction tools to help perform diagnosis. Having a confidence set would both help the doctor estimate the confidence of the prediction (i.e., smaller confidence sets imply higher confidence), but also give a sense of the set of possible diagnoses. Second, having a confidence set can be useful for reasoning about safety since they contain the true outcome with high probability. For instance, robots may use a confidence set over predicted trajectories to determine whether it is safe to act with high probability. As a concrete example, consider a self-driving car that uses a deep neural network to predict the path that a pedestrian might take. We require that the self-driving car avoid the pedestrian with high probability, which it can do by avoiding all possible paths in the predicted confidence set.

**Contributions.** We propose an algorithm combining calibrated prediction and statistical learning theory to construct PAC confidence sets for deep neural networks (Section 3). We propose instantiations of this framework in the settings of classification, regression, and learning models for reinforcement learning (Section 3.6). Finally, we evaluate our approach on three benchmarks: ResNet (He et al., 2016) for ImageNet (Russakovsky et al., 2015), a model (Held et al., 2016) learned for a visual object tracking benchmark (Wu et al., 2013), and a probabilistic dynamics model (Chua et al., 2018) learned for the half-cheetah environment (Brockman et al., 2016) (Section 4). Examples of ImageNet images with different sized ResNet confidence sets are shown in Table 1. As can be seen, our confidence sets become larger and the images become more challenging to classify. In addition, we show predicted confidence sets for ResNet in Table 2, as well as predicted confidence sets for the visual object tracking model in Table 3.

**Related work.** There has been work on constructing confidence sets with theoretical guarantees. Oftentimes, these guarantees are asymptotic rather than finite sample (Steinberger & Leeb, 2016; 2018). Alternatively, there has been work focused on predicting confidence sets with a given expected size (Denis & Hebiri, 2017).

More relatedly, there has been recent work on obtaining PAC guarantees. For example, there has been some work specific prediction tasks such as binary classification (Lei, 2014; Wang & Qiao,

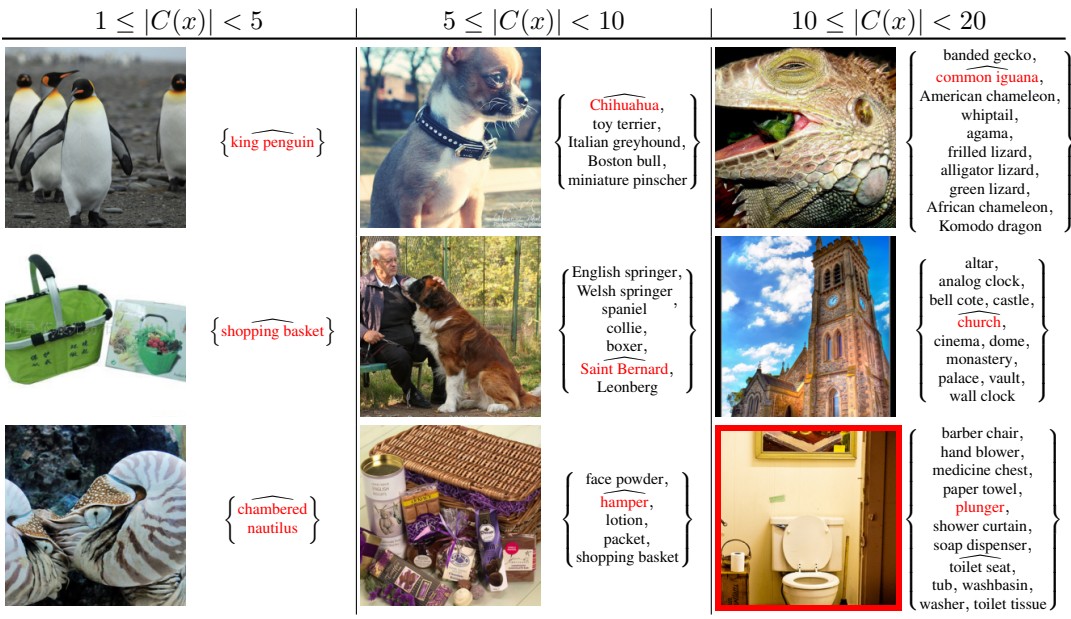

Table 2: Confidence sets of ImageNet images with varying ResNet confidence set sizes. The predicted confidence set is shown to the right of the corresponding input image. The true label is shown in red, and the predicted label is shown with a hat. See Table 5 in Appendix D for more examples.

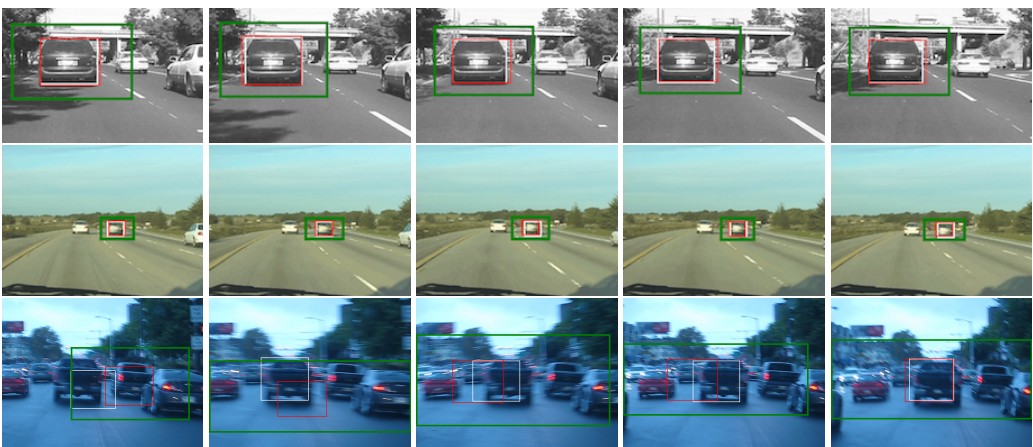

Table 3: Visualization of confidence sets for the tracking dataset (Wu et al., 2013), including the ground truth bounding box (white), the bounding box predicted by the original neural network (Held et al., 2016) (red), and the bounding box produced using our confidence set predictor (green). We have overapproximated the predicted ellipsoid confidence set with a box. Our bounding box contains the ground truth bounding box with high probability. See Table 9 in Appendix D for more examples.

2018). There has also been work in the setting of regression (Lei et al., 2018; Barber et al., 2019). However, in this case, the confidence sets are fixed in size—i.e., they do not depend on the input $x$ (Barber et al., 2019). Furthermore, they make stability assumptions about the learning algorithm (though they achieved improved rates by doing so) (Lei et al., 2018; Barber et al., 2019).

The most closely related work is on *conformal prediction* (Papadopoulos, 2008; Vovk, 2013). Like our approach, this line of work provides a way to construct confidence sets from a given confidence predictor, and provided PAC guarantees for the validity of these confidence sets. Indeed, with some work, our generalization bound Theorem 1 can be shown to be equivalent to Theorem 1 in Vovk (2013). In contrast to their approach, we proposed to use calibrated prediction to construct confidence predictors that can suitably be used with deep neural networks. Furthermore, our approach

makes explicit the connections to temperature scaling and as well as to generalization bounds from statistical learning theory (Vapnik, 1999). In addition, unlike our paper, they do not explicitly provide an efficient algorithm for constructing confidence sets. Finally, we also propose an extension to the case of learning models for reinforcement learning.

Finally, we build on a long line of work on *calibrated prediction*, which aims to construct "calibrated" probabilities (Murphy, 1972; DeGroot & Fienberg, 1983; Platt, 1999; Zadrozny & Elkan, 2001; 2002; Naeini et al., 2015; Kuleshov & Liang, 2015). Roughly speaking, probabilities are calibrated if events happen at rates equal to the predicted probabilities. This work has recently been applied to obtaining confidence estimates for deep neural networks (Guo et al., 2017; Kuleshov et al., 2018; Pearce et al., 2018), including for learned models for reinforcement learning (Malik et al., 2019). However, these approaches do not come with PAC guarantees.

## 2 PAC CONFIDENCE SETS

Our goal is to estimate confidence sets that are as small as possible, while simultaneously ensuring that they are *probably approximately correct (PAC)* (Valiant, 1984). Essentially, a confidence set is correct if it contains the true label. More precisely, let $\mathcal{X}$ be the inputs and $\mathcal{Y}$ be the labels, and let $D$ be a distribution over $\mathcal{Z} = \mathcal{X} \times \mathcal{Y}$. A *confidence set predictor* is a function $C : \mathcal{X} \to 2^{\mathcal{Y}}$ such that $C(x) \subseteq \mathcal{Y}$ is a set of labels; we denote the set of all confidence set predictors by $\mathcal{C}$. For a given example $(x, y) \sim D$, we say $C$ is *correct* if $y \in C(x)$. Then, the error of $C$ is

$$L(C) = \mathbb{P}_{(x,y)\sim D}[y \notin C(x)]. \tag{1}$$

Finally, consider an algorithm $\mathcal{A}$ that takes as input a validation set $Z_{\mathrm{val}} \subseteq \mathcal{Z}$ consisting of $n$ i.i.d. samples $(x, y) \sim D$, and outputs a confidence set predictor $\hat{C}$. Given $\epsilon, \delta \in \mathbb{R}_{>0}$, we say that $\mathcal{A}$ is *probably approximately correct (PAC)* if

$$\mathbb{P}_{Z_{\mathrm{val}}\sim D^n}\left[L(\hat{C}) > \epsilon \text{ where } \hat{C} = \mathcal{A}(Z_{\mathrm{val}})\right] < \delta. \tag{2}$$

Our goal is to design an algorithm $\mathcal{A}$ that satisfies (2) while constructing confidence sets $C(x)$ that are as "small in size" as possible on average. The size of $C(x)$ depends on the domain. For classification, we consider confidence sets that are arbitrary subsets of labels $C(x) \subseteq \mathcal{Y} = \{1, ..., Y\}$, and we measure the size by $|C(x)| \in \mathbb{N}$—i.e., the number of labels in $C(x)$. For regression, we consider confidence sets that are intervals $C(x) = [a, b] \subseteq \mathcal{Y} = \mathbb{R}$, and we measure size by $b - a$—i.e., the length of the predicted interval. Note that there is an intrinsic tradeoff between satisfying (2) and average size of $C(x)$—larger confidence sets are more likely to satisfy (2).

## 3 PAC ALGORITHM FOR CONFIDENCE SET CONSTRUCTION

Our algorithm is formulated in the *empirical risk framework*. Typically, this framework refers to *empirical risk minimization*. In our setting, such an algorithm would take as input (i) a parametric family of confidence set predictors $\mathcal{C} = \{C_\theta \mid \theta \in \Theta\}$, where $\Theta$ is the parameter space, and (ii) a training set $Z_{\mathrm{val}} \subseteq \mathcal{Z}$ of $n$ i.i.d. samples $(x, y) \sim D$, and output the confidence set predictor $C_{\hat{\theta}}$, where $\hat{\theta}$ minimizes the empirical risk:

$$\hat{\theta} = \underset{\theta\in\Theta}{\arg\min}\, \hat{L}(C_\theta; Z_{\mathrm{val}}) \qquad \text{where} \qquad \hat{L}(C; Z_{\mathrm{val}}) = \frac{1}{n} \sum_{(x,y)\in Z_{\mathrm{val}}} \mathbb{I}[y \notin C(x)].$$

Here, $\mathbb{I}[\phi] \in \{0, 1\}$ is the indicator function, and the empirical risk $\hat{L}$ in an estimate of the confidence set error (1) based on the validation set $Z_{\mathrm{val}}$.

However, our algorithm does not minimize the empirical risk. Rather, recall that our goal is to minimize the size of the predicted confidence sets given a PAC constraint on the true risk $L(\hat{\theta})$ based on the given PAC parameters $\epsilon, \delta \in \mathbb{R}_{>0}$ and the number of available validation samples $n = |Z_{\mathrm{val}}|$. Thus, the risk shows up as a constraint in the optimization problem, and the objective is instead to minimize the size of the predicted confidence sets:

$$\hat{\theta} = \underset{\theta\in\Theta}{\arg\min}\, S(\theta) \quad \text{subj. to} \quad \hat{L}(C_\theta; Z_{\mathrm{val}}) \leq \alpha. \tag{3}$$

At a high level, the value $\alpha = \alpha(n, \epsilon, \delta) \in \mathbb{R}_{\geq 0}$ is chosen to enforce the PAC constraint, and is based on generalization bounds from statistical learning theory (Valiant, 1984). Furthermore, following the temperature scaling approach (Platt, 1999), the parameter space $\Theta$ is chosen to be as small as possible (in particular, one dimensional) to enable good generalization. Finally, our choice of size metric $S$ follows straightforwardly based on our choice of parameter space. In the remainder of this section, we describe the choices of (i) parameter space $\Theta$, (ii) size metric $S(\theta)$, and (iii) confidence level $\alpha(n, \epsilon, \delta)$ in more detail, as well as how to solve (3) given these choices.

## 3.1 CHOICE OF PARAMETER SPACE $\Theta$

**Probability forecasters.** Our construction of the parameteric family of confidence set predictors $C_\theta$ assumes given a *probability forecaster* $f : \mathcal{X} \to \mathcal{P}_{\mathcal{Y}}$, where $\mathcal{P}_{\mathcal{Y}}$ is a space of probability distributions over $\mathcal{Y}$. Given such an $f$, we use $f(y \mid x)$ to denote the probability of label $y$ under distribution $f(x)$. Intuitively, $f(y \mid x)$ should be the probability (or probability density) that $y$ is the true label for a given input $x$—i.e., $f(y \mid x) \approx \mathbb{P}_{(X,Y) \sim D}[Y = y \mid X = x]$. For example, in classification, we can choose $\mathcal{P}_{\mathcal{Y}}$ to be the space of categorical distributions over $\mathcal{Y}$, and $f$ may be a neural network whose last layer is a softmax layer with $|\mathcal{Y}|$ outputs. Then, $f(y \mid x) = f(x)_y$. Alternatively, in regression, we can choose $\mathcal{P}_{\mathcal{Y}}$ to be the space of Gaussian distributions, and $f$ may be a neural network whose last layer outputs the values $(\mu, \sigma) \in \mathbb{R} \times \mathbb{R}_{>0}$ of a Gaussian distribution. Then, $f(y \mid x) = \mathcal{N}(x; \mu(x), \sigma(x)^2)$, where $(\mu(x), \sigma(x)) = f(x)$, and $\mathcal{N}(\cdot; \mu, \sigma^2)$ is the Gaussian density function with mean $\mu$ and variance $\sigma^2$.

**Training a probability forecaster.** To train a probability forecaster, we use a standard approach to calibrated prediction that combines maximum likelihood estimation with *temperature scaling*. [2] First, we consider a parametric model family $\mathcal{F} = \{f_\phi \mid \phi \in \Phi\}$, where $\Phi$ is the parameter space. Note that $\Phi$ can be high-dimensional—e.g., the weights of a neural network model. Given a training set $Z_{\text{train}} \subseteq \mathcal{Z}$ of $m$ i.i.d. samples $(x, y) \sim D$, the maximum likelihood estimate (MLE) of $\phi$ is

$$\hat{\phi} = \arg\min_{\phi \in \Phi} \ell(\phi; Z_{\text{train}}) \qquad \text{where} \qquad \ell(\phi; Z_{\text{train}}) = - \sum_{(x,y) \in Z_{\text{train}}} \log f_\phi(y \mid x). \qquad (4)$$

We could now use $f_{\hat{\phi}}$ as the probability forecaster. However, the problem with directly using $\hat{\phi}$ is that because $\hat{\phi}$ may be high-dimensional, it often overfits the training data $Z_{\text{train}}$. Thus, the probabilities are typically overconfident compared to what they should be.

To reduce their confidence, we use the *temperature scaling* approach to *calibrate* the predicted probabilities (Platt, 1999; Guo et al., 2017). Intuitively, this approach is to train an MLE estimate using exactly the same approach used to train $\hat{\phi}$, but using a single new parameter $\tau \in \mathbb{R}_{>0}$. The key idea is that this time, the model family is based on the parameters $\hat{\phi}$ from (4). In other words, the "shape" of the probabilities forecast by $f_{\hat{\phi}}$ are preserved, but their exact values are shifted.

More precisely, consider the model family $\mathcal{F}' = \{f_{\hat{\phi}, \tau} \mid \tau \in \mathbb{R}_{>0}\}$, where

$$f_{\hat{\phi}, \tau}(y \mid x) \propto \exp\left( \tau \log f_{\hat{\phi}}(y \mid x) \right).$$

Then, we have the following MLE for $\tau$:

$$\hat{\tau} = \arg\min_{\tau \in \mathbb{R}_{>0}} \ell'(\tau; Z'_{\text{train}}) \qquad \text{where} \qquad \ell'(\tau; Z'_{\text{train}}) = - \sum_{(x,y) \in Z'_{\text{train}}} \log f_{\hat{\phi}, \tau}(y \mid x). \qquad (5)$$

Note that $\hat{\tau}$ is estimated based on a second training set $Z'_{\text{train}}$. Because we are only fitting a single parameter, this training set can be much smaller than the training set $Z_{\text{train}}$ used to fit $\hat{\phi}$.

**Parametric family of confidence set predictors.** Finally, given a probability forecaster $f$, we consider one dimensional parameter space $\Theta = \mathbb{R}$; in an analogy to the temperature scaling technique for calibrated prediction, we denote this parameter by $T \in \Theta$. In particular, we assume a confidence probability predictor $f$ is given, and consider

$$C_T(x) = \{y \in \mathcal{Y} \mid f(y \mid x) \geq e^{-T}\}.$$

---

[2] A priori, it is not obvious that using temperature scaling can improve our confidence set predictor; we give a detailed discussion in Appendix A.1.

In other words, $C_T(x)$ is the set of $y$ with high probability given $x$ according to $f$. Considering this scalar parameter space, we denote the minimum of (3) by $\hat{T}$.

## 3.2 Choice of Size Metric $S(T)$

To choose the size metric $S(T)$, we note that for our chosen parametric family of confidence set predictors, smaller values correspond to uniformly smaller confidence sets—i.e.,

$$T \leq T' \Rightarrow \forall x, \ C_T(x) \subseteq C_{T'}(x).$$

Thus, we can simply choose the size metric to be

$$S(T) = T. \tag{6}$$

This choice minimizes the size of the confidence sets produced by our algorithm.

## 3.3 Choice of Confidence Level $\alpha(n, \epsilon, \delta)$

**Naive approach based on VC generalization bound.** A naive approach to choosing $\alpha(n, \epsilon, \delta)$ is to do so based on the VC dimension generalization bound (Vapnik, 1999). It is not hard to show that the problem of estimating $\hat{T}$ is equivalent to a binary classification problem, and that the VC dimension of $\Theta$ for this problem is 1. Thus, the VC dimension bound implies that for all $T \in \Theta$,

$$\mathbb{P}_{Z_{\text{val}} \sim D^n} \left[ L(C_T) \leq \hat{L}(C_T; Z_{\text{val}}) + \sqrt{\frac{\log(2n) + 1 - \log(\delta/4)}{n}} \right] \geq 1 - \delta. \tag{7}$$

The details of this equivalence are given in Appendix B.2. Then, suppose we choose

$$\alpha(n, \epsilon, \delta) = \epsilon - \sqrt{\frac{\log(2n) + 1 - \log(\delta/4)}{n}}.$$

With this choice, for the solution $\hat{T}$ of (3) with $\alpha = \alpha(n, \epsilon, \delta)$, the constraint in (3) ensures that $\hat{L}(C_{\hat{T}}; Z_{\text{val}}) \leq \alpha(n, \epsilon, \delta)$. Together with the VC generalization bound (7), we have

$$\mathbb{P}_{Z_{\text{val}} \sim D^n} \left[ L(C_{\hat{T}}) > \epsilon \right] < \delta,$$

which is exactly desired the PAC constraint on our predicted confidence sets.

**Direct generalization bound.** In fact, we can get better choices of $\alpha$ by directly bounding generalization error. For instance, in the realizable setting (i.e., we always have $\hat{L}(C_{\hat{T}}; Z_{\text{val}}) = 0$), we can get rates of $n = \tilde{O}(1/\epsilon)$ instead of $n = \tilde{O}(1/\epsilon^2)$ (Kearns & Vazirani, 1994); see Appendix A.2 for details. We can achieve these rates by choosing $\alpha = 0$, but then, the PAC guarantees we obtain may actually be stronger than desired (i.e., for $\epsilon' < \epsilon$). Intuitively, we can directly prove a bound that interpolates between the realizable setting and the VC generalization bound—in particular:

**Theorem 1.** *For any $\epsilon \in [0, 1]$, $n \in \mathbb{N}_{>0}$, and $k \in \{0, 1, ..., n\}$, we have*

$$\mathbb{P}_{Z_{val} \sim D^n} \left[ L(C_{\hat{T}}) > \epsilon \right] \leq \sum_{i=0}^{k} \binom{n}{i} \epsilon^i (1 - \epsilon)^{n-i},$$

*where $\hat{T}$ is the solution to (3) with $\alpha = k/n$.* [3]

We give a proof in Appendix B.2. Based on Theorem 1, we can choose

$$\alpha(n, \epsilon, \delta) = \max_{k \in \mathbb{N} \cup \{0\}} k/n \ \ \text{subj. to} \ \ \sum_{i=0}^{k} \binom{n}{i} \epsilon^i (1 - \epsilon)^{n-i} < \delta. \tag{8}$$

## 3.4 Theoretical Guarantees

We have the following guarantee, which follows straightforwardly from Theorem 1:

**Corollary 1.** *Let $\hat{T}$ be the solution to (3) for $\alpha = \alpha(n, \epsilon, \delta)$ chosen according to (8). Then, we have*

$$\mathbb{P}_{Z_{val} \sim D^n} \left[ L(C_{\hat{T}}) > \epsilon \right] < \delta.$$

In other words, our algorithm is probably approximately correct.

---

[3]The theorem statement relies on additional standard technical conditions; see Appendix B.1.

---

**Algorithm 1** Algorithm for solving (3).

---

**procedure** ESTIMATECONFIDENCESETPREDICTOR($Z_{\text{train}}, Z'_{\text{train}}, Z_{\text{val}}$)
    Estimate $\hat{\phi}, \hat{\tau}$ using (4) and (5), respectively
    Compute $\alpha(n, \epsilon, \delta)$ according to (8) by enumerating $k \in \{0, 1, ..., n\}$
    Let $k^* = n \cdot \alpha(n, \epsilon, \delta)$ (note that $k \in \{0, 1, ..., n\}$)
    Sort $(x, y) \in Z_{\text{val}}$ in ascending order of $f_{\hat{\phi}, \hat{\tau}}(y \mid x)$
    Let $(x_{k^*+1}, y_{k^*+1})$ be the $(k^* + 1)$st element in the sorted $Z_{\text{val}}$
    Solve (3) by choosing $\hat{T} = -\log f_{\hat{\phi}, \hat{\tau}}(y_{k^*+1} \mid x_{k^*+1})$
    Return $C_{\hat{T}} : x \mapsto \{y \in \mathcal{Y} \mid f_{\hat{\phi}, \hat{\tau}}(y \mid x) \geq e^{-\hat{T}}\}$
**end procedure**

---

## 3.5 PRACTICAL IMPLEMENTATION

Our algorithm for estimating a confidence set predictor $C_{\hat{T}}$ is summarized in Algorithm 1. The algorithm solves the optimization problem (3) using the choices of $\Theta$, $S(T)$, and $\alpha(n, \epsilon, \delta)$ described in the preceding sections. There are two key implementation details that we describe here.

**Computing $\alpha(n, \epsilon, \delta)$.** To compute $\alpha(n, \epsilon, \delta)$, we need to solve (8). A straightforward approach is to enumerate all possible choices of $k \in \{0, 1, ..., n\}$. There are two optimizations. First, the objective is monotone increasing in $k$, so we can enumerate $k$ in ascending order until the constraint no longer holds. Second, rather than re-compute the left-hand side of the constraint $\sum_{i=0}^{k} \binom{n}{i} \epsilon^i (1 - \epsilon)^{n-i}$, we can accumulate the sum as we iterate over $k$. We can also incrementally compute $\binom{n}{i}$, $\epsilon^i$, and $(1 - \epsilon)^{n-i}$. For numerical stability, we perform these computations in log space.

**Solving (3).** To solve (3), note that the constraint in (3) is equivalent to

$$\sum_{(x,y) \in Z_{\text{val}}} E(x, y; T) \leq n \cdot \alpha(n, \epsilon, \delta) \quad \text{where} \quad E(x, y; T) = \mathbb{I}\left[f_{\hat{\phi}, \hat{\tau}}(y \mid x) < e^{-T}\right]. \quad (9)$$

Also, note that $k^* = n \cdot \alpha(n, \epsilon, \delta)$ is an integer due to the definition of $\alpha(n, \epsilon, \delta)$ in (8). Thus, we can interpret (9) as saying that $E(x, y; T) = 1$ for at most $k^*$ of the points $(x, y) \in Z_{\text{val}}$.

In addition, note that $E(x, y; T)$ decreases monotonically as $f_{\hat{\phi}, \hat{\tau}}(y \mid x)$ becomes larger. Thus, we can sort the points $(x, y) \in Z_{\text{val}}$ in ascending order of $f_{\hat{\phi}, \hat{\tau}}(y \mid x)$, and require that only the first $k^*$ points $(x, y)$ in this list satisfy $E(x, y; T) = 1$. In particular, letting $(x_{k^*+1}, y_{k^*+1})$ be the $(k^* + 1)$st point, (9) is equivalent to

$$f_{\hat{\phi}, \hat{\tau}}(y_{k^*+1} \mid x_{k^*+1}) \geq e^{-T}. \quad (10)$$

In other words, this constraint says that $T$ must satisfy $y_{k^*+1} \in C_T(x_{k^*+1})$. Finally, the solution $\hat{T}$ to (3) is the smallest $T$ that satisfies (10), which is the $T$ that makes (10) hold with equality—i.e.,

$$\hat{T} = -\log f_{\hat{\phi}, \hat{\tau}}(y_{k^*+1} \mid x_{k^*+1}). \quad (11)$$

We have assumed $f_{\hat{\phi}, \hat{\tau}}(y_{k^*+1} \mid x_{k^*+1}) > f_{\hat{\phi}, \hat{\tau}}(y_{k^*} \mid x_{k^*})$; if not, we decrement $k^*$ until this holds.

## 3.6 PROBABILITY FORECASTERS FOR SPECIFIC TASKS

We briefly discuss the architectures we use for probability forecasters for various tasks. We give details, including how we measure the sizes of predicted confidence sets $C_T(x)$, in Appendix C. We consider three tasks: classification, regression, and model-based reinforcement learning. For classification, we use the standard approach of using a soft-max layer to predict label probabilities $f(y \mid x)$. For regression, we also use a standard approach where the neural network predicts both the mean $\mu(x)$ and covariance $\Sigma(x)$ of a Gaussian distribution $\mathcal{N}(\mu(x), \Sigma(x))$; then, $f(y \mid x) = \mathcal{N}(y; \mu(x), \Sigma(x))$ is the probability density of $y$ according to this Gaussian distribution.

Finally, for model-based reinforcement learning, our goal is to construct confidence sets over trajectories predicted using a learned model of the dynamics. We consider unknown dynamics

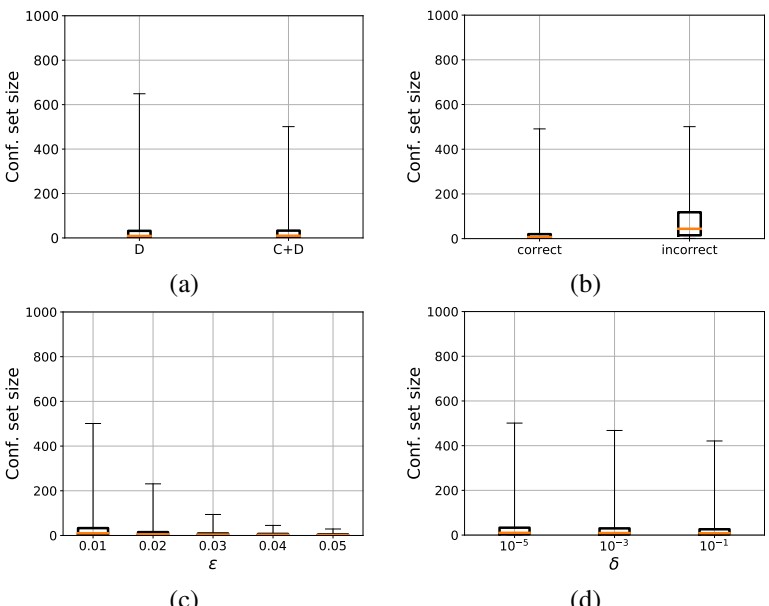

Figure 1: Results on ResNet for ImageNet with $n = 20000$. Default parameters are $\epsilon = 0.01$ and $\delta = 10^{-5}$. We plot the median and min/max confidence set sizes. (a) Ablation study; $C$ is "calibrated predictor" (i.e., use $f_{\hat{\phi},\hat{\tau}}$ instead of $f_{\hat{\phi}}$), and $D$ is "direct bound" (i.e., use Theorem 1 instead of the VC generalization bound). (b) Restricted to correctly vs. incorrectly labeled images. (c) Varying $\epsilon$. (d) Varying $\delta$.

$g^*(x' \mid x, u)$ mapping a state-action pair $(x, u)$ to a distribution over states $x'$, and consider a known (and fixed) policy $\pi(u \mid x)$ mapping a given state $x$ to a distribution over actions $u \in \mathcal{U} \subseteq \mathbb{R}^{d_U}$. Then, we let $f^*(x' \mid x) = \mathbb{E}_{\pi(u \mid x)}[g^*(x' \mid u)]$ denote the (unknown) closed-loop dynamics.

Next, we consider a forecaster $f(x' \mid x) \approx f^*(x' \mid x)$ of the form $f(x' \mid x) = \mathcal{N}(x'; \mu(x), \Sigma(x))$, and our goal is to construct confidence sets for the predictions of $f$. However, we want to do so for not just for one-step predictions, but for predictions over a time horizon $H \in \mathbb{N}$. In particular, given initial state $x_0 \in \mathcal{X}$, we can sample $x^*_{1:H} = (x_1, ..., x_H) \sim f^*$ by letting $x^*_0 = x_0$ and sequentially sampling $x^*_{t+1} \sim f(\,\cdot\, \mid x^*_t)$ for each $t \in \{0, 1, ..., H-1\}$. Then, our goal is to construct a confidence set that contains $x^*_{1:H} \in \mathcal{X}^H$ with high probability (over both the randomness in an initial state distribution $x_0 \sim d_0$ and the randomness in $f^*$).

To do so, we construct and use a forecaster $\tilde{f}(x_{1:H} \mid x_0)$ based on $f$. In principle, this task is a special case of multivariate regression, where the inputs are $\mathcal{X}$ (i.e., the initial state $x_0$) and the outputs are $\mathcal{Y} = \mathcal{X}^H$ (i.e., a predicted trajectory $x_{1:H}$). However, the variance $\Sigma(x)$ predicted by our probability forecaster is only for a single step, and does not take into account the fact that $x$ is itself uncertain. Thus, we use a simple heuristic where we accumulate variances over time. More precisely, we construct (i) the predicted mean $\bar{x}_{1:H} = (\bar{x}_1, ..., \bar{x}_H)$ by $\bar{x}_0 = x_0$ and $\bar{x}_{t+1} = \mu(\bar{x}_t)$ for $t \in \{0, 1, ..., H-1\}$, and (ii) the predicted variances $\tilde{\Sigma}_{1:H} = (\tilde{\Sigma}_1, ..., \tilde{\Sigma}_H)$ by

$$\tilde{\Sigma}_t = \Sigma(\bar{x}_0) + \Sigma(\bar{x}_1) + ... + \Sigma(\bar{x}_{t-1}).$$

We use a probability forecaster $\tilde{f}(x_{1:H} \mid x_0) = \mathcal{N}(x_{1:H}; \bar{x}_{1:H}, \tilde{\Sigma}_{1:H})$ to construct confidence sets.

## 4 EXPERIMENTS

We describe our experiments on ImageNet (a classification task), a visual object tracking benchmark (a regression task), and the half-cheetah environment (a model-based reinforcement learning task). We give additional results in Appendix D.

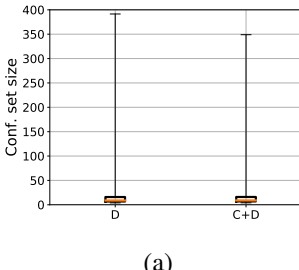 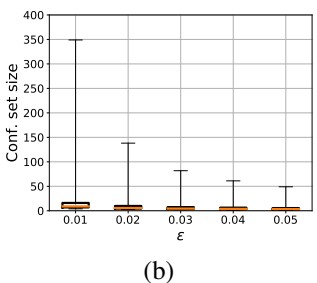 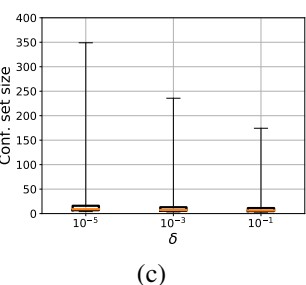

(a)           (b)           (c)

Figure 2: Confidence set sizes for an object tracking benchmark (Wu et al., 2013); we use $n = 5,000$, $\epsilon = 0.01$, and $\delta = 10^{-5}$. (a) Ablation study similar to Figure 3. In (b) and (c), we show how the confidence set sizes produced using our algorithm vary with respect to $\epsilon$ and $\delta$, respectively.

**ResNet for ImageNet.** We use our algorithm to compute confidence sets for ResNet (He et al., 2016) on ImageNet (Russakovsky et al., 2015), for $\epsilon = 0.01$, $\delta = 10^{-5}$, and $n = 20000$ validation images. We show the results in Figure 1. In (a), we compare our approach to an ablation. In particular, $C$ refers to performing an initial temperature scaling step to calibrate the neural network predictor (i.e., using $f_{\hat{\phi}}$ instead of $f_{\hat{\phi},\hat{\tau}}$), and (ii) $D$ refers to using Theorem 1 instead of the VC generalization bound. Thus, $C + D$ refers to our approach. As can be seen, using calibrated predictor produces a noticeable reduction in the maximum confidence set size.

We also compared to the ablation $C$—i.e., using the VC generalization bound. However, we were unable to obtain valid confidence sets for our choice of $\epsilon$ and $\delta$—i.e., (3) is infeasible. That is, using Theorem 1 outperforms using the VC generalization bound since the VC bound is too loose to satisfy the PAC criterion for our choice of parameters. In addition, in Table 6 in Appendix D, we show results for larger choices of $\epsilon$ and $\delta$; these results show that our approach substantially outperforms the ablation based on the VC bound even when the VC bound produces valid confidence sets.

In (b), we show the confidence set sizes for images correctly vs. incorrectly labeled by ResNet. As expected, the sizes are substantially larger for incorrectly labeled images. Finally, in (c) and (d), we show how the sizes vary with $\epsilon$ and $\delta$, respectively. As expected, the dependence on $\epsilon$ is much more pronounced (note that $\delta$ is log-scale).

**Visual object tracking.** We apply our confidence set prediction algorithm to a 2D visual single-object tracking task, which is a multivariate regression problem. Specifically, the input space $\mathcal{X}$ consists of the previous image, the previous bounding box (in $\mathbb{R}^4$), and the current image. The output space $\mathcal{Y} = \mathbb{R}^4$ is a current bounding box. We use the regression-based tracker from Held et al. (2016), and retrain the regressor neural network to predict the mean and variance of a Gaussian distribution. More precisely, our object tracking model predicts the mean and variance of each bounding box parameter—i.e., $(x_{\min}, y_{\min}, x_{\max}, y_{\max})$. Given this bounding box forecaster $f_{\hat{\phi}}$, we calibrate and estimate a confidence set predictor as described in Section 3.6.

We use the visual object tracking benchmark from Wu et al. (2013) to train and evaluate our confidence set predictor. This benchmark consists of 99 video sequences labeled with ground truth bounding boxes. We randomly split these sequences to form the training set for calibration, validation set for confidence set estimation, and test set for evaluation. For each sequence, a pair of two adjacent frames constitute a single example. Our training dataset contains 20,882 labeled examples, each consisting of of a pair of consecutive images and ground truth bounding boxes. The validation set for confidence set estimation and test set contain 22,761 and 22,761 labeled examples, respectively. Figure 2 shows the sizes of the predicted confidence sets; the sizes are measured as described in Section 3.6 for regression tasks. As for ResNet, we omit results for the VC bound ablation since $n$ is too small to get a bound. The trends are similar to the ones for ResNet.

**Half-cheetah.** We use our algorithm to compute confidence sets for a probabilistic neural network dynamics model (Chua et al., 2018) for the half-cheetah environment (Brockman et al., 2016), for $\epsilon = 0.01$, $\delta = 10^{-5}$, $H = 20$ time steps, and $n = 5000$ validation rollouts. When using temperature scaling to calibrate $f_{\hat{\phi}}$ to obtain $f_{\hat{\phi},\hat{\tau}}$, we calibrate each dimension of time steps independently (i.e., we fit $H$ parameters, where $H$ is time horizon). We show the results in Figure 3.

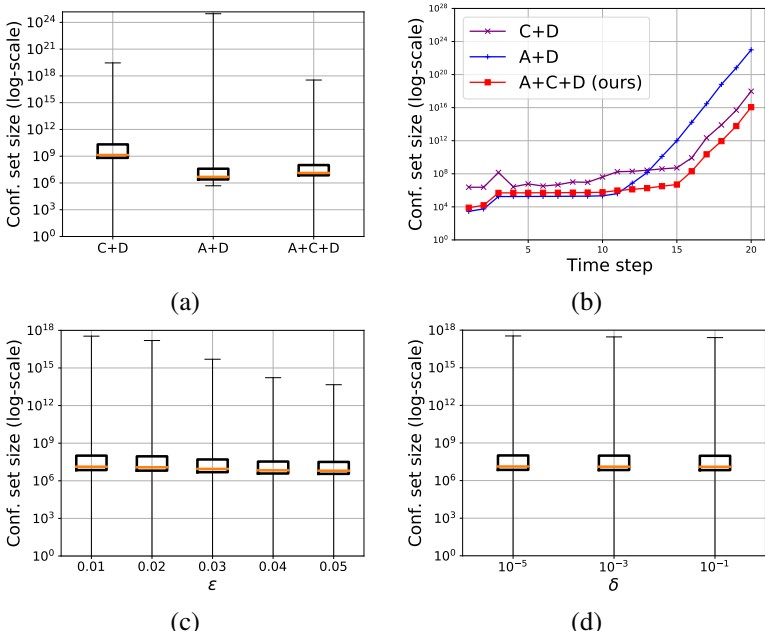

Figure 3: Results on the dynamics model for the half-cheetah with $n = 5000$. Default parameters are $\epsilon = 0.01$ and $\delta = 10^{-5}$. (a) Ablation study; $A$ is "accumulated variance" (i.e., for each $t \in \{1, ..., 20\}$, use $\tilde{\Sigma}_t$ instead of $\Sigma_t = \Sigma(\bar{x}_{t-1})$), and $C$ and $D$ are as for ResNet. We plot the median and min/max confidence set sizes (see Section 3.6), averaged across $t \in \{1, ..., 20\}$. (b) Same ablations, but with per time step size. We plot the average size of the confidence set for the predicted state $x_t$ on step $t$, as a function of $t \in \{1, ..., 20\}$. (c) Varying $\epsilon$, and (d) varying $\delta$.

In (a), we compare to two ablations. The labels $C$ and $D$ are as for ResNet; in addition, $A$ refers to using the accumulated variance $\tilde{\Sigma}_t$ instead of the one-step predicted variances $\Sigma_t = \Sigma(\bar{x}_{t-1})$. Thus, $A + C + D$ is our approach. As before, we omit results for the ablation using the VC generalization bound since $n$ is so small that the bound does not hold for any $k$ for the given $\epsilon$ and $\delta$. In (b), we show the same ablations over the entire trajectory until $t = 20$. As can be seen, using the calibrated predictor produces a large gain; these gains are most noticeable in the tails. Using the accumulated confidence produces a smaller, but still significant, gain. In (c) and (d), we show how the sizes vary with $\epsilon$ and $\delta$, respectively. The trends are similar those for ResNet.

## 5    CONCLUSION

We have proposed an algorithm for constructing PAC confidence sets for deep neural networks. Our approach leverages statistical learning theory to obtain theoretical guarantees on the predicted confidence sets. These confidence sets quantify the uncertainty of deep neural networks. For instance, they can be used to inform safety-critical decision-making, and to ensure safety with high-probability in robotics control settings that leverage deep neural networks for perception. Future work includes extending these results to more complex tasks (e.g., structured prediction), and handling covariate shift (e.g., to handle policy updates in reinforcement learning).

#### ACKNOWLEDGMENTS

This work was support in part by NSF CCF-1910769 and by the Air Force Research Laboratory and the Defense Advanced Research Projects Agency under Contract No. FA8750-18-C-0090. Any opinions, findings and conclusions or recommendations expressed in this material are those of the author(s) and do not necessarily reflect the views of the Air Force Research Laboratory (AFRL), the Defense Advanced Research Projects Agency (DARPA), the Department of Defense, or the United States Government.

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

# A    DISCUSSION OF ALGORITHM DESIGN CHOICES

## A.1    USEFULNESS OF TEMPERATURE SCALING

In this section, we discuss why temperature scaling can help improve the predicted confidence sets. A concern is that temperature scaling does not change the ordering of label probabilities. Thus, we may expect that temperature scaling does not affect the predicted confidence sets. However, this fact only holds when considering a single input $x$—i.e., the ordering of the probabilities $p(y \mid x)$ for $y \in \mathcal{Y}$ is not changed by temperature scaling. Indeed, the order of confidences for labels for different inputs can change. For a concrete example, consider two inputs $x$ and $x'$, and the case $\mathcal{Y} = \{0, 1, 2\}$. Assume that the label probabilities are

$$f( \cdot \mid x) = [1/3 \quad 1/3 \quad 1/3]^\top$$
$$f( \cdot \mid x') = [3/4 \quad 1/4 \quad 0]^\top.$$

Now, if we take temperature $\tau$ very large, then the labels become roughly

$$f_\tau( \cdot \mid x) = [1/3 \quad 1/3 \quad 1/3]^\top$$
$$f_\tau( \cdot \mid x') = [1/2 \quad 1/2 \quad 0]^\top.$$

As a consequence, there are confidence sets that are achievable when using $f_\tau$ that are not achievable when using $f$. In particular, the confidence sets

$$C_T(x) = \varnothing$$
$$C_T(x') = \{0, 1\}$$

can be achieved using $f_\tau$ (e.g., with $e^{-T} = 2/5$). However, it is impossible to achieve these confidence sets using $f$ for any choice of $T$, since if $1 \in C_T(x')$, then it must be the case that $C_T(x) = \{0, 1, 2\}$. Intuitively, we expect calibrated prediction to improve the ordering of probabilities across different inputs. Our experiments support this intuition, since they show that empirically, using calibrated predictors $f_\tau$ produces confidence sets of smaller size.

## A.2    USEFULNESS OF DIRECT BOUND

One key design choice is to use a specialized generalization bound that directly provides PAC guarantees on our confidence sets rather than simply applying the VC dimension bound. The easiest way to determine which bound is better is to examine which one produces a smaller confidence set. In our approach, the size of the confidence set decreases monotonically with the choice of $\alpha = \alpha(n, \epsilon, \delta)$ in (3). Thus, the bound that produces larger $\alpha$ is better. Recall that the VC dimension bound produces

$$\alpha_{\text{VC}}(n, \epsilon, \delta) = \epsilon - \sqrt{\frac{\log(2n) + 1 - \log(\delta/4)}{n}},$$

whereas our direct bound produces (for $k = 0$)

$$\alpha_{\text{direct}}(n, \epsilon, \delta) = \max_{k \in \mathbb{N}} k/n \quad \text{subj. to} \quad \sum_{i=0}^{k} \binom{n}{i} \epsilon^i (1 - \epsilon)^{n-i} < \delta.$$

Directly comparing these two choices of $\alpha$ is difficult, but our experiments show empirically that using the direct bound outperforms using the indirect bound.

A more direct way to compare the two approaches is to instead ask how large $n$ needs to be to achieve $\alpha(n, \epsilon, \delta) = 0$. For $\alpha_{\text{VC}}$, it is easy to check that we need

$$n \geq \frac{\log(2n) + 1 + \log(4/\delta)}{\epsilon^2}.$$

Thus, we need $n$ to be *at least* $O(\log(1/\delta)/\epsilon^2)$ (and possibly greater, to account for the $\log(2n)$ term). In contrast, for our direct bound, $\alpha = 0$ corresponds to the case $k = 0$. To achieve $k = 0$, it suffices to have $n$ satisfying $(1 - \epsilon)^n < \delta$. Using $(1 - \epsilon)^n \leq e^{-n\epsilon}$, it suffices to have $n$ satisfying

$$n \geq \frac{\log(1/\delta)}{\epsilon}.$$

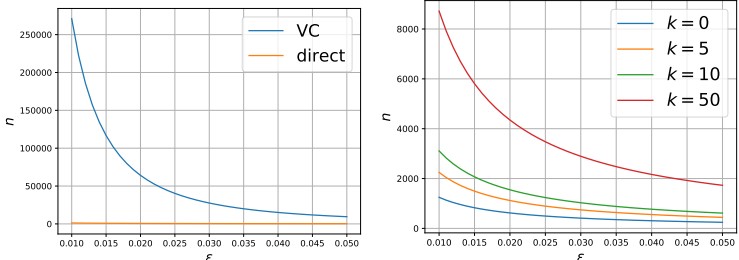

Figure 4: Sample complexity of different bounds; we fix $\delta = 10^{-5}$. Left: Sample complexity of VC bound and direct bound when $k = 0$. Right: Sample complexity of direct bound for varying $k$.

In other words, $n$ only needs to be $O(\log(1/\delta)/\epsilon)$. For small $\epsilon$ (e.g., $\epsilon = 0.01$), we need $100\times$ fewer samples to achieve the same size confidence set (i.e., with choice $\alpha(n, \epsilon, \delta) = 0$). In Figure 4 (right), we compute the exact values of $n$ needed to get $\alpha(n, \epsilon, \delta) = 0$ as a function of $\epsilon$ for each bound (fixing $\delta = 10^{-5}$). As expected, our bound requires substantially smaller $n$.

Finally, in Figure 4 (right), we compare the magnitude of $n$ needed to achieve larger values of $\alpha$ using our direct bound; for simplicity, we actually consider larger values of $k$ (where $\alpha = k/n$), but the qualitative insights are the same. As can be seen, even for large $k$, (e.g., $k = 50$), the number of samples increases, but not substantially.

# B  THEORETICAL GUARANTEES

## B.1  ASSUMPTIONS

We make two additional technical assumptions in Theorem 1, both of which are standard. First, we assume that $f$ is measurable; this assumption holds for all models used in practice, including neural networks (e.g., it holds as long as $f$ is continuous).

Second, letting $\phi : \mathcal{Z} \to \mathbb{R}$, where $\mathcal{Z} = \mathcal{X} \times \mathcal{Y}$, be defined by $\phi((x, y)) = -\log f(y \mid x)$, we assume that the distribution $\bar{D}$ induced by $\phi$ on $\mathbb{R}$ has continuous cumulative distribution function (CDF). More precisely, letting $\mu_D$ be the measure defining $D$, then $\bar{D}$ is defined by the measure

$$\mu_{\bar{D}}(t) = \mu_D(\phi^{-1}(t)),$$

where $\phi^{-1} : \mathbb{R} \to 2^{\mathcal{Z}}$ is the inverse of $\phi$ in the sense that $z \in \phi^{-1}(\phi(z))$ for all $z \in \mathcal{Z}$. Then, we assume that the CDF corresponding to $\bar{D}$ is continuous. This second assumption is standard in statistical learning theory (Kearns & Vazirani, 1994). Essentially, it says that for any $t \in \mathbb{R}$, the probability that $t = -\log f(y \mid x)$ must equal zero. This assumption should hold unless $p(x, y)$ or $f(y \mid x)$ are degenerate in some way. Furthermore, we can detect this case. In particular, the failure mode corresponds to the case that we see multiple points with the same value $-\log f(y \mid x)$. Thus, choosing $\hat{T} = -\log f(y \mid x)$ would include all these points, so the realized error rate $\alpha$ is larger than desired for $\hat{T}$. In this case, we can simply choose a slightly larger $\hat{T}$ to avoid this problem.

## B.2  PROOF OF THEOREM 1

At a high level, our proof proceeds in three steps. First, we show that a confidence set predictor $C_T$ can be encoded as a binary classifier $M_T$. Second, we show that a PAC bound for $M_T$ implies a PAC bound for $C_T$ (where in both cases, the unknown parameter is $T \in \mathbb{R}$). Third, we prove PAC bounds on the error of $M_{\hat{T}}$; by the second step, these bounds complete our proof.

**Encoding $C_T$ as a binary classifier $M_T$.** We begin by showing how the problem of learning a PAC confidence set predictor $C_T$ reduces to the problem of learning a PAC binary classifier $M_T$. First, we show that for any $T \in \mathbb{R}$, the confidence set predictor $C_T$ can be encoded as a binary classifier $M_T$. Consider any parameter $T \in \Theta = \mathbb{R}$. Recall that we use the model $f(y \mid x)$ to construct the confidence set predictor

$$C_T(x) = \{y \in \mathcal{Y} \mid f(y \mid x) \geq e^{-T}\}.$$

Now, define the map $\phi : \mathcal{Z} \to \mathbb{R}$ by $\phi(x, y) = -\log f(y \mid x)$, where $\mathcal{Z} = \mathcal{X} \times \mathcal{Y}$, and define the binary classifier $M_T : \mathbb{R} \to \{0, 1\}$ by

$$M_T(t) = \mathbb{I}[t \leq T].$$

Here, $\mathbb{I}[s]$ is the indicator function, which returns one if a statement $s$ is true and zero otherwise. We claim that

$$C_T(x) = \{y \in \mathcal{Y} \mid M_T(\phi(x, y)) = 1\}. \tag{12}$$

To see this claim, note that

$$\begin{aligned}
C_T(x) &= \{y \in \mathcal{Y} \mid f(y \mid x) \geq e^{-T}\} \\
&= \{y \in \mathcal{Y} \mid -\log f(y \mid x) \leq T\} \\
&= \{y \in \mathcal{Y} \mid \phi(x, y) \leq T\} \\
&= \{y \in \mathcal{Y} \mid \mathbb{I}[\phi(x, y) \leq T] = 1\} \\
&= \{y \in \mathcal{Y} \mid M_T(\phi(x, y)) = 1\},
\end{aligned}$$

as claimed.

**PAC bound for $M_T$ implies PAC bound for $C_T$.** Next, we show that a PAC bound for $M_T$ implies a PAC bound for $C_T$. More precisely, we design a data distribution $\tilde{D}$ and loss $\tilde{\ell}$, and show that (i) the distribution of $\tilde{T}$ (trained to optimize $M_T$) is the same as the distribution of $\hat{T}$ (constructed using our algorithm), and (ii) a PAC bound for $M_{\tilde{T}}$ (where $\tilde{T}$ is trained on data from $\tilde{D}$) implies a PAC bound for $C_{\tilde{T}}$. We show that as a consequence, a PAC bound on $M_{\tilde{T}}$ implies a PAC bound on $C_{\hat{T}}$.

We begin by constructing $\tilde{D}$ and $\tilde{\ell}$. To this end, recall that $D$ is a given distribution over $\mathcal{X} \times \mathcal{Y}$. We define a data distribution $\tilde{D}$ over $\tilde{\mathcal{X}} \times \tilde{\mathcal{Y}}$, where $\tilde{\mathcal{X}} = \mathbb{R}$ and $\tilde{\mathcal{Y}} = \{0, 1\}$, as follows. The first component of $\tilde{D}$ is the distribution over $\tilde{\mathcal{X}}$ induced by $\phi$ from $D$, and the second component is the distribution over $\tilde{\mathcal{Y}}$ that places all probability mass on 1. Formally, $\tilde{D}$ exists assuming $\phi$ is measurable, so the induced distribution exists; for all our choices of $f$ (i.e., categorical or Gaussian), this property is satisfied. Then,

$$\mu_{\tilde{D}}((t, a)) = \mu_D(\phi^{-1}(t)) \cdot \mathbb{I}[a = 1],$$

where $\mu_{\tilde{D}}$ is the measure encoding $\tilde{D}$, and $\mu_D$ is the measure encoding $D$. Furthermore, we define $\ell : \tilde{\mathcal{Y}} \times \tilde{\mathcal{Y}} \to \{0, 1\}$ to be the 0-1 loss $\ell(a, a') = \mathbb{I}[a \neq a']$. Finally, let $\hat{T}$ be chosen using our algorithm—i.e.,

$$\hat{T} = \arg\min T \quad \text{subj. to} \quad L(C_T; Z) \leq \alpha$$

$$L(C_T; Z) = \frac{1}{|Z|} \sum_{(x,y) \in Z} \mathbb{I}[y \notin C_T(x)],$$

for any $\alpha \in \mathbb{R}_{\geq 0}$, and let $\tilde{T}$ be chosen similarly for $M_T$—i.e.,

$$\tilde{T} = \arg\min T \quad \text{subj. to} \quad L(M_T; \tilde{Z}) \leq \alpha$$

$$L(M_T; \tilde{Z}) = \frac{1}{|\tilde{Z}|} \sum_{(t,a) \in \tilde{Z}} \ell(M_T(t), a) = \frac{1}{|\tilde{Z}|} \sum_{(t,a) \in \tilde{Z}} \mathbb{I}[M_T(t) \neq a].$$

Now, we show (i) above. In particular, we claim that $\hat{T}(Z)$ has the same distribution as $\tilde{T}(\tilde{Z})$, where $Z \sim D^n$ and $\tilde{Z} \sim \tilde{D}^n$ are random datasets. To this end, define $\Phi : \mathcal{Z}^n \mapsto \tilde{\mathcal{Z}}^n$ by

$$\Phi((z_1, ..., z_n)) = ((\phi(z_1), 1), ..., (\phi(z_n), 1)).$$

Note that

$$\begin{aligned}
\tilde{L}(M_T; \Phi(Z)) &= \frac{1}{|\Phi(Z)|} \sum_{i=1}^{n} \mathbb{I}[M_T(\phi(x_i, y_i)) \neq 1] \\
&= \frac{1}{|Z|} \sum_{i=1}^{n} \mathbb{I}[y_i \notin C_T(x_i)] \\
&= L(C_T; Z),
\end{aligned}$$

from which it follows that

$$\hat{T}(Z) = \arg\min \ T \quad \text{subj. to} \quad L(C_T; Z) \le \alpha$$
$$= \arg\min \ T \quad \text{subj. to} \quad \tilde{L}(M_T; \Phi(Z)) \le \alpha$$
$$= \tilde{T}(\Phi(Z)).$$

By construction of $\Phi$, the random variables $\tilde{Z}$ and $\Phi(Z)$ have the same distribution; thus, it follows that the random variables $\tilde{T}(\tilde{Z})$ and $\tilde{T}(\Phi(Z))$ have the same distribution as well. Since $\hat{T}(Z) = \tilde{T}(\Phi(Z))$, it follows that $\hat{T}(Z)$ has the same distribution as $\tilde{T}(\tilde{Z})$, as claimed.

Next, we show (ii) above. In particular, we claim that a PAC bound for $M_{\tilde{T}(\tilde{Z})}$—i.e.,

$$\mathbb{P}_{\tilde{Z} \sim \tilde{D}^n}[\tilde{L}(M_{\tilde{T}(\tilde{Z})}) \le \epsilon] \ge 1 - \delta,$$

implies a PAC bound for $C_{\tilde{T}(\tilde{Z})}$—i.e.,

$$\mathbb{P}_{\tilde{Z} \sim \tilde{D}^n}[L(C_{\tilde{T}(\tilde{Z})}) \le \epsilon] \ge 1 - \delta,$$

where the true losses are

$$\tilde{L}(M_T) = \mathbb{E}_{(t,a) \sim \tilde{D}}[\ell(M_T(t), a)] = \mathbb{P}_{(t,a) \sim \tilde{D}}[M_T(t) \ne a]$$
$$L(C_T) = \mathbb{E}_{(x,y) \sim D}[\mathbb{I}[y \notin C_T(x)]] = \mathbb{P}_{(x,y) \sim D}[y \notin C_T(x)].$$

Note that it suffices to show that the true loss for $C_T$ equals the true loss for $M_T$—i.e.,

$$L(C_T) = \tilde{L}(M_T),$$

since this equation (together with the PAC bound for $M_{\tilde{T}(\tilde{Z})}$) implies

$$\mathbb{P}_{\tilde{Z} \sim \tilde{D}^n}[L(C_{\tilde{T}(\tilde{Z})}) \le \epsilon] = \mathbb{P}_{\tilde{Z} \sim \tilde{D}^n}[\tilde{L}(M_{\tilde{T}(\tilde{Z})}) \le \epsilon] \ge 1 - \delta,$$

as desired. To see the claim, note that

$$\tilde{L}(M_T) = \mathbb{P}_{(t,a) \sim \tilde{D}}[M_T(t) \ne a]$$
$$= \int \mathbb{I}[M_T(t) \ne a] d\mu_{\tilde{D}}((t,a))$$
$$= \sum_{a=0}^{1} \mathbb{I}[a = 1] \cdot \int \mathbb{I}[M_T(t) \ne a] d\mu_D(\phi^{-1}(t))$$
$$= \int \mathbb{I}[M_T(t) \ne 1] d\mu_D(\phi^{-1}(t))$$

Now, using the change of variables $t \mapsto \phi(z)$, we have

$$\tilde{L}(M_T) = \int \mathbb{I}[M_T(\phi(z)) \ne 1] d\mu_D(z)$$
$$= \int \mathbb{I}[M_T(\phi(x,y)) \ne 1] \cdot D(x,y) dx dy.$$

Then, using (12), we have

$$\tilde{L}(M_T) = \int \mathbb{I}[y \notin C_T(x)] D(x,y) dx dy$$
$$= \mathbb{P}_{(x,y) \sim D}[y \notin C_T(x)]$$
$$= L(C_T),$$

as claimed.

Finally, combining (i) and (ii), we have

$$\mathbb{P}_{Z \sim D^n}[L(C_{\hat{T}(Z)}) \le \epsilon] = \mathbb{P}_{\tilde{Z} \sim \tilde{D}^n}[L(C_{\tilde{T}(\tilde{Z})}) \le \epsilon] \ge 1 - \delta,$$

where the first equality follows since (i) says that $\hat{T}(Z)$ (where $Z \sim D^n$) has the same distribution as $\tilde{T}(\tilde{Z})$ (where $\tilde{Z} \sim \tilde{D}^n$), and the second inequality follows by (ii).

**Generalization bound.** Finally, we prove the PAC bound

$$\mathbb{P}_{\tilde{Z} \sim \tilde{D}^n}[\tilde{L}(M_{\hat{T}}) \leq \epsilon] \geq 1 - \delta_0, \tag{13}$$

for $M_{\tilde{T}}$, where $\delta_0 = \sum_{i=0}^{k} \binom{n}{i} \epsilon^i (1 - \epsilon)^{n-i}$; for conciseness, we have dropped the dependence of $\tilde{T}$ on $\tilde{Z}$. By the previous step, this bound implies the theorem statement. To this end, we first simplify the left-hand side of the inequality (13). In particular, let $T^*$ be the smallest $T$ for which $\tilde{L}(M_{T^*}) = \epsilon$; such a $T^*$ exists by our assumption that $\tilde{D}$ has continuous density function.

First, we claim that $T < T^*$ implies $\tilde{L}(M_T) > \tilde{L}(M_{T^*})$. Assuming $T < T^*$, then

$$\begin{aligned}
\tilde{L}(M_T) &= \mathbb{P}_{(t,a) \sim \tilde{D}}[M_T(t) \neq a] \\
&= \mathbb{E}_{(t,a) \sim \tilde{D}}[\mathbb{I}[M_T(t) \neq a] \\
&= \mathbb{E}_{(t,a) \sim \tilde{D}}[\mathbb{I}[M_{\hat{T}}(t) \neq 1]] \\
&= \mathbb{E}_{(t,a) \sim \tilde{D}}[\mathbb{I}[\mathbb{I}[t \leq T] \neq 1]] \\
&= \mathbb{E}_{(t,a) \sim \tilde{D}}[\mathbb{I}[t > T]] \\
&> \mathbb{E}_{(t,a) \sim \tilde{D}}[\mathbb{I}[t > T^*]] \\
&= \tilde{L}(M_{T^*}).
\end{aligned}$$

Assuming $T \geq T^*$, we can similarly show that $\tilde{L}(M_{\hat{T}}) \leq \tilde{L}(M_{T^*})$. It follows that

$$\begin{aligned}
\mathbb{P}_{\tilde{Z} \sim \tilde{D}^n}\left[\tilde{L}(M_{\hat{T}}) > \epsilon\right] &= \mathbb{P}_{\tilde{Z} \sim \tilde{D}^n}\left[\tilde{L}(M_{\hat{T}}) > \tilde{L}(M_{T^*})\right] \\
&= \mathbb{P}_{\tilde{Z} \sim \tilde{D}^n}\left[\tilde{T} < T^*\right].
\end{aligned}$$

As a consequence, (13) is equivalent to

$$\mathbb{P}_{\tilde{Z} \sim \tilde{D}^n}\left[\tilde{T} < T^*\right] \leq \delta_0.$$

Next, recall that $\tilde{T}$ must satisfy $\tilde{L}(M_{\tilde{T}}; \tilde{Z}) \leq \alpha$, where

$$\tilde{L}(M_{\tilde{T}}; \tilde{Z}) = \frac{1}{n} \sum_{(t,a) \in \tilde{Z}} \mathbb{I}[M_{\tilde{T}}(t) \neq a].$$

Assuming $\hat{T} < T^*$, and using $k = n \cdot \alpha$, it follows that

$$\begin{aligned}
k \geq \sum_{(t,a) \in \tilde{Z}} \mathbb{I}[M_{\tilde{T}}(t) \neq a] &= \sum_{(t,a) \in \tilde{Z}} \mathbb{I}[M_{\tilde{T}}(t) \neq 1] \\
&= \sum_{(t,a) \in \tilde{Z}} \mathbb{I}[t > \tilde{T}] \\
&\geq \sum_{(t,a) \in \tilde{Z}} \mathbb{I}[t > T^*].
\end{aligned}$$

As a consequence, we have

$$\begin{aligned}
\mathbb{P}_{\tilde{Z} \sim \tilde{D}^n}\left[\tilde{T} < T^*\right] &\leq \mathbb{P}_{\tilde{Z} \sim \tilde{D}^n}\left[\sum_{(t,a) \in \tilde{Z}} \mathbb{I}[t > T^*] \leq k\right] \\
&= \sum_{i=0}^{k} \mathbb{P}_{\tilde{Z} \sim \tilde{D}^n}\left[\sum_{(t,a) \in \tilde{Z}_{\text{val}}} \mathbb{I}[t > T^*] = i\right].
\end{aligned}$$

By our definition of $T^*$, the event in the final expression says that the sum of $n$ i.i.d. Bernoulli random variables $\mathbb{I}[t > T^*] \sim \text{Bernoulli}(\epsilon)$ is at most $k$. Thus, this event follows a distribution $\text{Binomial}(n, \epsilon)$, so

$$\mathbb{P}_{\tilde{Z} \sim \tilde{D}^n}\left[\tilde{T} < T^*\right] \leq \sum_{i=0}^{k} \text{Binomial}(i; n, \epsilon) = \sum_{i=0}^{k} \binom{n}{i} \epsilon^i (1-\epsilon)^{n-i} = \delta_0,$$

as claimed. The theorem statement follows. ∎

## C   DETAILS ON PROBABILITY FORECASTERS FOR SPECIFIC TASKS

In this section, we describe architectures for probability forecasters for classification, regression, and model-based reinforcement learning.

**Classification.** For the case $\mathcal{Y} = \{1, ..., Y\}$, we choose the probability forecaster $f$ to be a neural network with a softmax output. Then, we can compute a given confidence set

$$C_T(x) = \{y \in \mathcal{Y} \mid f(y \mid x) \geq e^{-T}\}$$

by explicitly enumerating $y \in \mathcal{Y}$. We measure the size of $C_T(x)$ as $|C_T(x)|$.

**Regression.** For the case $\mathcal{Y} = \mathbb{R}$, we choose the probability forecaster $f$ to be a neural network that outputs the parameters $(\mu, \sigma) \in \mathcal{Y} \times \mathbb{R}_{>0}$ of a Gaussian distribution. Then, we have

$$C_T(x) = \left[\mu - \sigma\sqrt{2(T - \log(\sigma\sqrt{2\pi}))}, \mu + \sigma\sqrt{2(T - \log(\sigma\sqrt{2\pi}))}\right].$$

This choice generalizes to $\mathcal{Y} = \mathbb{R}^d$ by having $f$ output the parameters $(\mu, \Sigma) \in \mathcal{Y} \times \mathbb{S}_{\succ 0}^d$ (where $\mathbb{S}_{\succ 0}^d$ is the set of $d$ dimensional symmetric positive definite matrices) of a $d$ dimensional Gaussian distribution. Note that $C_T(x)$ is an ellipsoid $C_T(x) = \mu + \Lambda S^{d-1}$, where $\Lambda \in \mathbb{R}^{d \times d}$ and $S^{d-1}$ is the unit sphere in $\mathbb{R}^d$; in particular, $\Lambda = D^{-\frac{1}{2}}Q$, where $QDQ^\top$ is the eigendecomposition of

$$(2T - d\ln 2\pi - \ln\det\Sigma)^{-1} \cdot \Sigma^{-1}.$$

We measure the size of $C_T(x)$ as $\|\Lambda\|_F$, where $\|\cdot\|_F$ is the Frobenius norm.

**Model-based reinforcement learning.** In model-based reinforcement learning, the goal is to predict trajectories based on a model of the dynamics. We consider an MDP with states $\mathcal{X} \subseteq \mathbb{R}^{d_X}$, actions $\mathcal{U} \subseteq \mathbb{R}^{d_U}$, an unknown distribution over initial states $x_0 \sim d_0$, and unknown dynamics $g^*(x' \mid x, u)$ mapping a state-action pair $(x, u) \in \mathcal{X} \times \mathcal{U}$ to a distribution over states $x' \in \mathcal{X}$. We assume a fixed, known policy $\pi(u \mid x)$, mapping a state $x \in \mathcal{X}$ to a distribution over actions $u \in \mathcal{U}$. The (unknown) closed-loop dynamics are $f^*(x' \mid x) = \mathbb{E}_{\pi(u|x)}[g^*(x' \mid x, u)]$.

Given initial state $x_0 \in \mathcal{X}$ and time horizon $H \in \mathbb{N}$, we can sample a trajectory $x_{1:H}^* = (x_1^*, ..., x_H^*) \sim f^*$ by setting $x_0^* = x_0$ and sequentially sampling $x_{t+1}^* \sim f^*(\cdot \mid x_t^*)$ for $t \in \{0, 1, ..., H-1\}$. Our goal is to predict a confidence set $C_T(x_0) \subseteq \mathcal{X}^H$ that contains $x_{1:H}^* \in \mathcal{X}^H$ with high probability (according to both the randomness in initial states $x_0 \sim d_0$ and in $f$). This problem is a multivariate regression problem with inputs $\mathcal{X}$ and outputs $\mathcal{Y} = \mathcal{X}^H$.

We assume given a probability forecaster $f(x' \mid x) = \mathcal{N}(x'; \mu(x), \Sigma(x))$ trained to predict the distribution over next states—i.e., $f(x' \mid x) \approx f^*(x' \mid x)$. Given initial state $x_0 \in \mathcal{X}$ and time horizon $H \in \mathbb{N}$, we construct the mean trajectory $\bar{x}_{1:H}$ by setting $\bar{x}_0 = x_0$ and letting $\bar{x}_{t+1} = \mu(\bar{x}_t)$. To account for the fact that the variances accumulate over time, we sum them together to obtain the predicted variances $\tilde{\Sigma}_{1:H}$—i.e.,

$$\tilde{\Sigma}_t = \Sigma(\bar{x}_0) + \Sigma(\bar{x}_1) + ... + \Sigma(\bar{x}_{t-1}).$$

Then, we use the probability forecast $\tilde{f}(\bar{x}_{1:H}, \tilde{\Sigma}_{1:H}) = \mathcal{N}(\bar{x}_{1:H}, \tilde{\Sigma}_{1:H})$ (where we think of $\bar{x}_{1:H}$ as a vector in $\mathbb{R}^{H \cdot d_X}$ and $\tilde{\Sigma}_{1:H}$ as a block diagonal matrix in $\mathbb{R}^{(H \cdot d_X) \times (H \cdot d_X)}$) to construct confidence sets.

Finally, we describe how we measure the size of a predicted confidence set $C_T(x_0) \subseteq \mathcal{X}^H$. In particular, note that $C_T(x_0)$ has the form

$$C_T(x_0) = (C_{T,1}(x_0), ..., C_{T,H}(x_0)),$$

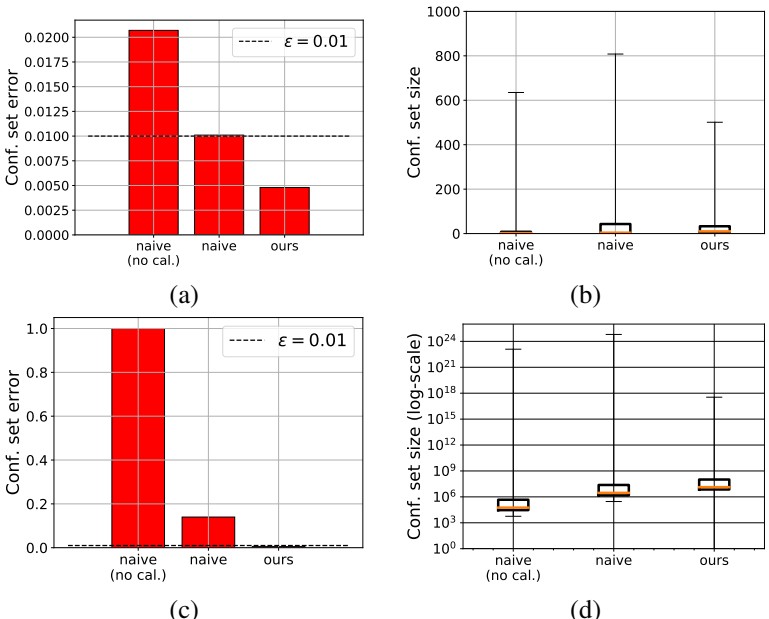

Figure 5: Comparison to baselines that do not have theoretical guarantees. In (a) and (b), we show results for ImageNet, and in (c) and (d), we show results for the half-cheetah. In (a) and (c), we show the empirical error in the confidence set sizes; the dotted line denotes $\epsilon = 0.01$, our target confidence set error. In (b) and (d), we show the sizes of the constructed confidence sets.

i.e., $C_{T,t}(x_0)$ is the confidence set for the state $x_t$ reached after $t$ time steps. Then, we measure the size of the confidence set for each component $C_{T,t}(x_0)$ (for $t \in \{1, ..., H\}$) individually, and take the average. As in the case of regression, $C_{T,t}(x_0)$ is an ellipsoid $C_{T,t}(x_0) = \bar{x}_t + \Lambda_t S^{d_X - 1}$; then, the size of $C_T(x_0)$ is $H^{-1} \sum_{t=1}^{H} \|\Lambda_t\|_F$.

An additional detail is that when we calibrate this forecaster, we calibrate each component $C_{T,t}(x_0)$ individually—i.e., we use $H$ calibration parameters $\tau_1, ..., \tau_H$.

## D   ADDITIONAL RESULTS

### D.1   COMPARISON TO ADDITIONAL BASELINES

We compare to two baselines that do not have theoretical guarantees. We assume given a probability forecaster $f(y \mid x)$. Then, given an input $x \in \mathcal{X}$, we construct the confidence set to satisfy

$$\sum_{y \in C(x)} f(y \mid x) \geq 1 - \epsilon. \tag{14}$$

More precisely, we first rank the labels in decreasing order of $f(y \mid x)$, to obtain a list $(y_1, y_2, ..., y_{|\mathcal{Y}|})$. Then, we choose the smallest $k$ such that (14) holds for $C(x) = \{y_1, ..., y_k\}$. Intuitively, if the probabilities $f(y \mid x)$ are correct (i.e., $f(y \mid x)$ is the true probability of $y$ given $x$), then this confidence set should contain the true label $y$ with high probability.

For regression, we cannot explicitly rank labels $y \in \mathcal{Y} \subseteq \mathbb{R}^d$, but they are monotonically decreasing away from the mean. Then, assuming $f(y \mid x) = \mathcal{N}(y; \mu(x), \Sigma(x))$ is Gaussian, we take an ellipsoid of shape $\Sigma(x)$ around $\mu(x)$ with minimum radius that captures $1 - \epsilon$ of the probability mass of $f(y \mid x)$. More precisely, we choose

$$C(x) = C_{\hat{T}(x)}(x)$$

$$\hat{T}(x) = \arg\min_{T \in \mathbb{R}} T \quad \text{subj. to} \quad \mathbb{P}_{f(y|x)}[y \in C_T(x)] \geq 1 - \epsilon,$$

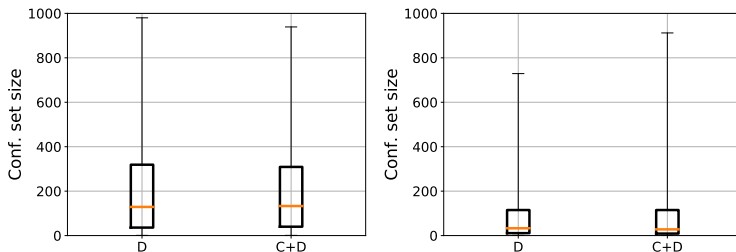

Figure 6: Confidence set sizes for two neural network architectures trained on ImageNet; for both, we use $n = 20,000$, $\epsilon = 0.01$ and $\delta = 10^{-5}$. Left: AlexNet (Krizhevsky, 2014); here, the empirical confidence set error of our approach $C + D$ is 0.0066. Right: GoogLeNet (Szegedy et al., 2015); here, the empirical confidence set error of our approach is 0.0061.

where $C_T(x) = \{y \in \mathcal{Y} \mid f(y \mid x) \geq e^{-T}\}$ as before. Note that unlike our algorithm, the threshold $\hat{T}(x)$ is not a learned parameter, but is computed independently for each new input $x$. We can solve for $\hat{T}(x)$ efficiently by changing basis to convert $f(y \mid x)$ to a standard Gaussian distribution, and then using the error function to compute the cutoff that includes the desired probability mass.

In Figure 5, we compare the confidence sets constructed using this approach with (i) the forecaster $f_{\hat{\phi}}(y \mid x)$ without any calibration, and (ii) the calibrated forecaster $f_{\hat{\phi}, \hat{\tau}}(y \mid x)$. We plot both the confidence set sizes and the empirical error rates. For the latter, recall that a confidence set predictor $C$ is correct if $L(C) < \epsilon$, where $L(C)$ the true error rate. However, we cannot measure $L(C)$; instead, we approximate it on a held-out test set $Z_{\text{test}} \subseteq \mathcal{X} \times \mathcal{Y}$—i.e., $L(C) \approx \hat{L}(C; Z_{\text{test}})$, where

$$\hat{L}(C; Z_{\text{test}}) = \frac{1}{|Z_{\text{test}}|} \sum_{(x,y) \in Z_{\text{test}}} \mathbb{I}[y \notin C(x)].$$

Intuitively, $\hat{L}(C; Z_{\text{test}})$ is the fraction of inputs $(x, y) \in Z_{\text{test}}$ such that the predicted confidence set for $x$ does not contain $y$. We say a confidence set $C$ is *empirically valid* when $\hat{L}(C; Z_{\text{test}}) < \epsilon$. Recall that our algorithm guarantees correctness with probability at least $1 - \delta$, where $\delta = 10^{-5}$.

As can be seen, the baseline approaches are not empirically valid in all cases. In one case—namely, the baseline with the calibrated forecaster on ImageNet—the confidence sets are almost empirically valid. However, in this case, the confidence sets are much larger than those based on our approach, despite the fact that the error rate of our confidence sets are empirically valid. Thus, our algorithms outperform the baselines in all cases.

### D.2 Results on Additional ImageNet Neural Network Architectures

We apply our approach to two additional neural network architectures for ImageNet: AlexNet (Krizhevsky, 2014) and GoogLeNet (Szegedy et al., 2015). Our results are shown in Figure 6. As can be seen, calibration reduces the confidence set sizes for AlexNet, but actually increases the confidence set sizes for GoogleNet. Thus, both calibrated and uncalibrated models may need to be considered when constructing confidence set predictors. Also, we find that confidence set sizes are correlated with classification error—the test errors for AlexNet, GoogleNet, and ResNet are $47.83\%$, $29.41\%$, and $21.34\%$, respectively, and their confidence set sizes decrease in the same order.

### D.3 Results on Additional Classification Datasets

We apply our approach to three small classification datasets: an Arrhythmia detection dataset (Guvenir et al., 1997), a car evaluation dataset (Bohanec & Rajkovic, 1988), and a medical alarm dataset (Bonafide et al., 2017). The confidence set sizes are shown in Figure 7. We choose larger values of $\epsilon$ and $\delta$ since we cannot obtain confidence sets that satisfy the PAC criterion with smaller $\epsilon$ and $\delta$ when the number of validation examples $n$ is too small. For all three datasets, the empirical confidence set error is smaller than the specified error $\epsilon$; thus, the constructed confidence sets are empirically valid. For these datasets, the confidence set sizes of our approach $C + D$ and our approach without calibration $D$ are similar, most likely due to the small number of class labels.

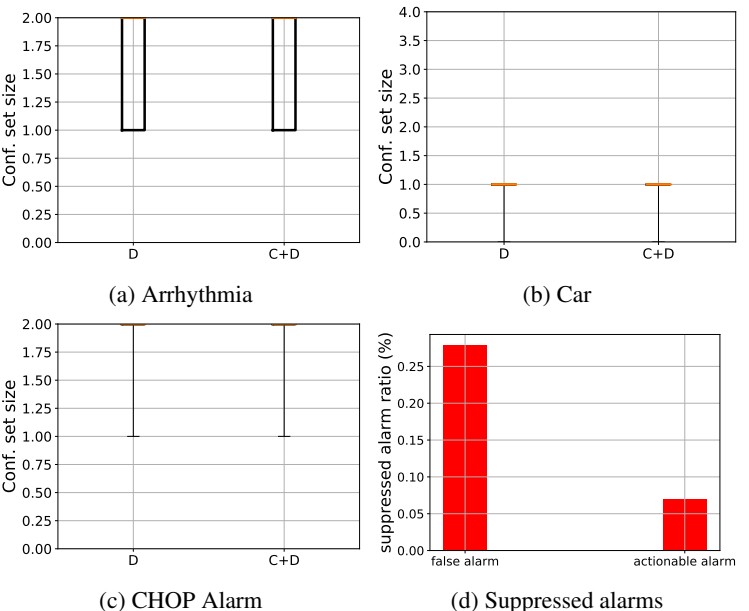

Figure 7: Confidence set sizes for three additional classification benchmarks: (a) the arrhythmia detection dataset (Guvenir et al., 1997); here, $n = 90$, $\epsilon = 0.1$, $\delta = 0.05$, and the empirical confidence set error of our approach $C + D$ is 0.0435, (b) the car evaluation dataset (Bohanec & Rajkovic, 1988); here, $n = 345$, $\epsilon = 0.05$, $\delta = 10^{-5}$, and the empirical confidence set error of our approach $C + D$ is 0.0172, and (c) the CHOP alarm dataset (Bonafide et al., 2017); here, $n = 1000$, $\epsilon = 0.02$, $\delta = 10^{-5}$, and the empirical confidence set error of our approach $C + D$ is 0.0159. (d) The fractions of actionable and false alarms with a confidence set $\{0\}$ (i.e., only contains false alarm).

We additionally ran our approach on a medical dataset where classification decisions are safety critical; thus, correct predicted confidence sets are required. In particular, we use the Children's Hospital of Philadelphia (CHOP) alarm dataset (Bonafide et al., 2017). This dataset consists of vital signs from 100 patients around one year of age. One of the vital signs is the oxygen level of the blood, and a medical device generates an alarm if the oxygen level is below a specified level. The labels indicate whether the generated alarm is true ($y = 1$) or false ($y = 0$). We use $n = 1000$, $\epsilon = 0.02$, and $\delta = 10^{-5}$. The empirical confidence set error of our approach is $\hat{L}(C; Z_{\text{test}}) = 0.0159$.

The key question is how many false alarms can be reliably detected using machine learning to help reduce alarm fatigue. We consider an approach where we use the predicted confidence sets to detect false alarms. In particular, we first train a probability forecaster $f : \mathcal{X} \to \mathcal{P}_{\mathcal{Y}}$, where $\mathcal{Y} = \{0, 1\}$, to predict the probability that an alarm is true, and then construct a calibrated confidence set predictor $\tilde{f} : \mathcal{X} \to 2^{\mathcal{Y}}$ based on this forecaster. We consider an alarm to be false if the predicted confidence set is $\tilde{f}(x) = \{0\}$—i.e., according to our confidence set predictor, the alarm is definitely false. Then, our PAC guarantee says that the alarm is actually false with probability at least $1 - \epsilon$. In summary, we suppress an alarm if $\tilde{f}(x) = \{0\}$. Using our approach, $176/630$ (i.e., 27.94%) of false alarms are suppressed, while only $13/187$ (i.e., 6.95%) true alarms are suppressed (see Figure 7 (d)).

### D.4 RESULTS ON ADDITIONAL REGRESSION DATASETS

We ran our algorithm on two small regression baselines—the Auto MPG dataset (Quinlan, 1993) and the student grade dataset (Cortez & Silva, 2008). We show results in Figure 8. The parameters we use are $\epsilon = 0.1$ and $\delta = 0.05$; as with the smaller classification datasets, we use larger choices of $\epsilon$ and $\delta$ since we cannot construct valid confidence sets for smaller choices. For the Auto MPG dataset, the empirical confidence set error of our final model $C + D$ is $\hat{L}(C; Z_{\text{test}}) = 0.0597$, so these are empirically valid. For the student grade dataset, the error is $\hat{L}(C; Z_{\text{test}}) = 0.1250$, which is slightly larger than desired; this failure is likely due to the fact that the failure probability $\delta = 0.05$ is somewhat large.

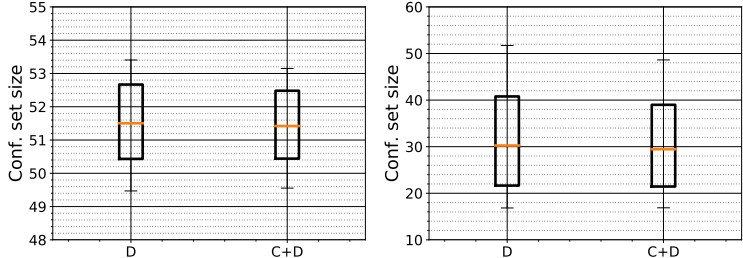

Figure 8: Confidence set sizes for two benchmarks focused on regression; for both, we use $\epsilon = 0.1$ and $\delta = 0.05$. Left: the Auto MPG dataset (Quinlan, 1993); here, $n = 70$, and the empirical confidence set error of our approach $C + D$ is $0.1250$. Right: The student grade dataset (Cortez & Silva, 2008); here, $n = 100$, and the empirical confidence set error of our approach is $0.0597$.

### D.5 Additional Results on ImageNet, Half-Cheetah, and Object Tracking

Table 4 & 5 show examples of ResNet confidence set sizes for ImageNet images. Table 6 shows results for varying $\epsilon, \delta$ on ResNet. Tables 7 & 8 show results for varying $\epsilon, \delta$ on the Half-Cheetah. Table 9 shows visualizations of the confidence sets predicted for our object tracking benchmark.

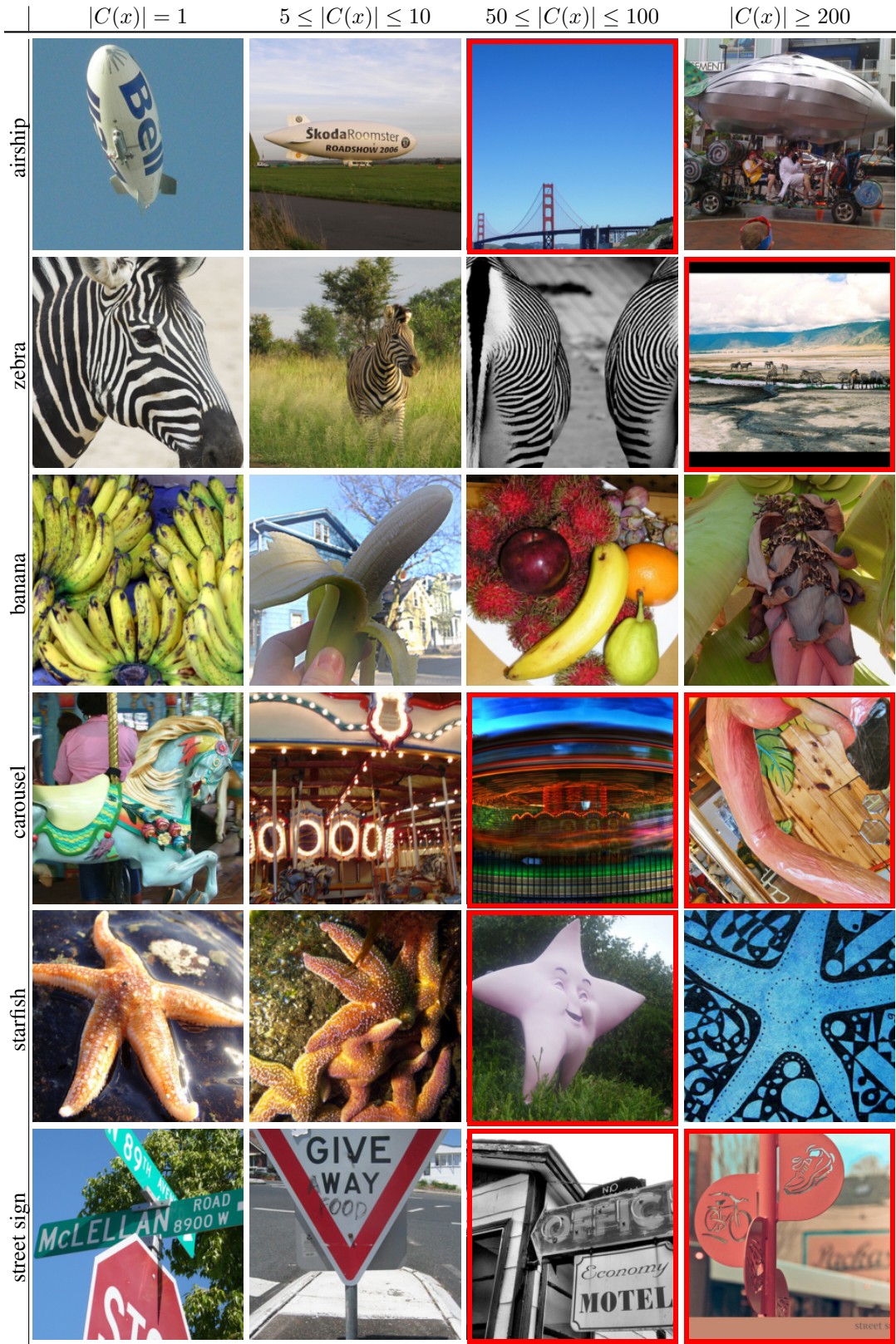

Table 4: ImageNet images with varying ResNet confidence set sizes. The confidence set sizes are on the top. The true label is on the left-hand side. Incorrectly labeled images are boxed in red.

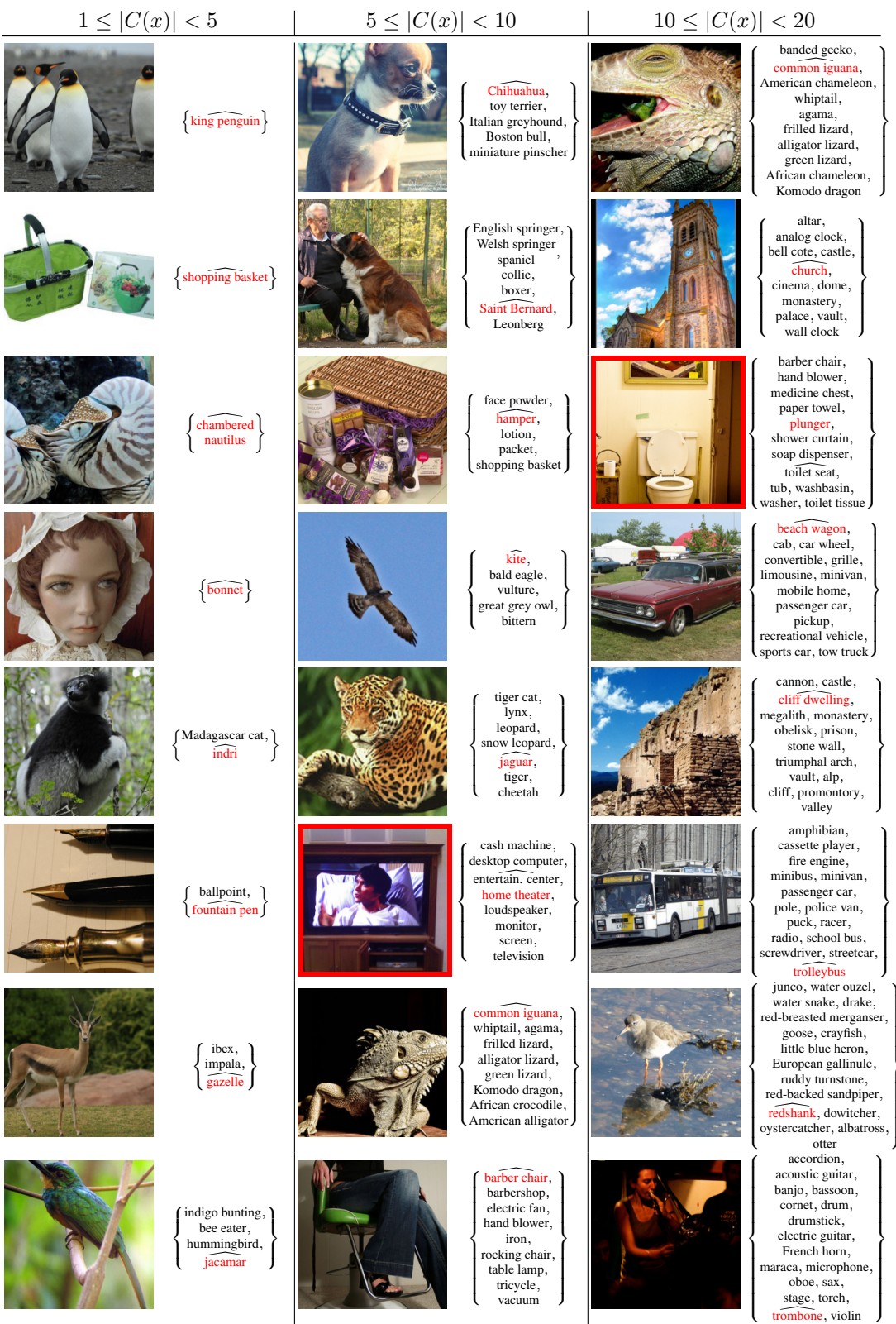

Table 5: Confidence sets of ImageNet images with varying ResNet confidence set sizes. The predicted confidence set is shown to the right of the corresponding input image. The true label is shown in red, and the predicted label is shown with a hat.

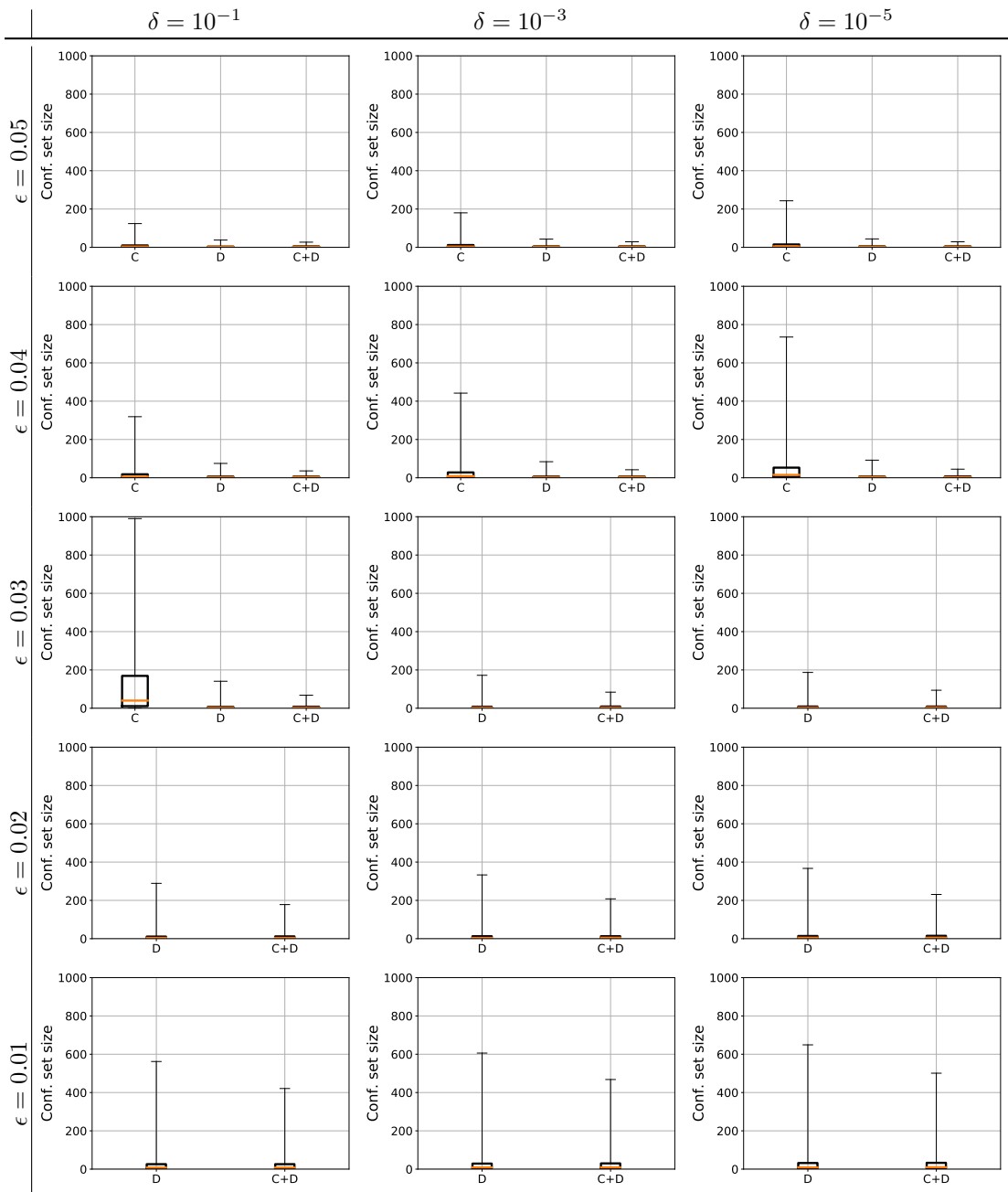

Table 6: Confidence set sizes for ResNet trained on ImageNet, for varying $\epsilon, \delta$ and for $n = 20,000$. The plots are as in Figure 1 (a).

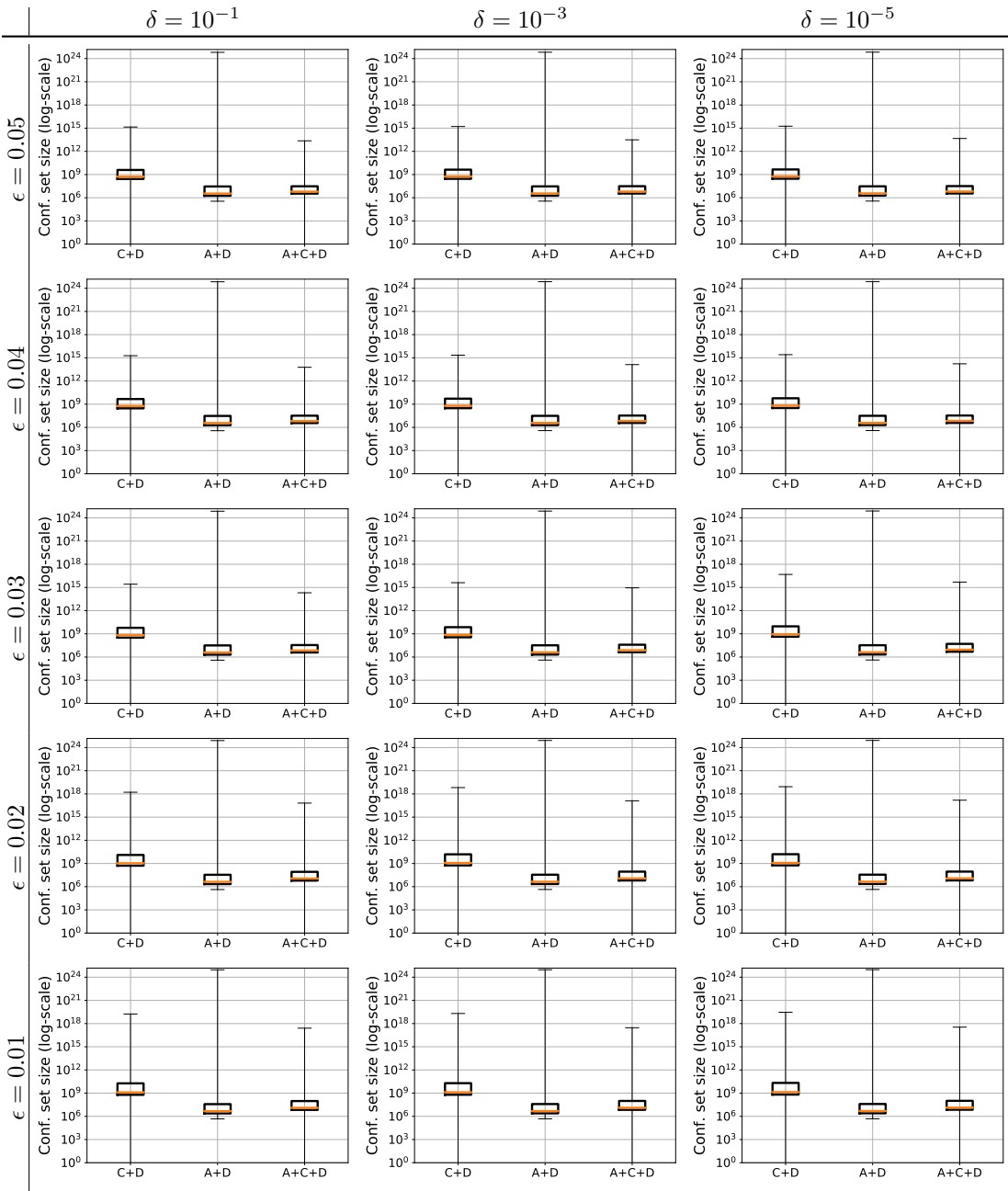

Table 7: Confidence set sizes for a neural network dynamics model trained on the half-cheetah environment, for varying $\epsilon, \delta$ and for $n = 5000$. The plots are as in Figure 3 (a).

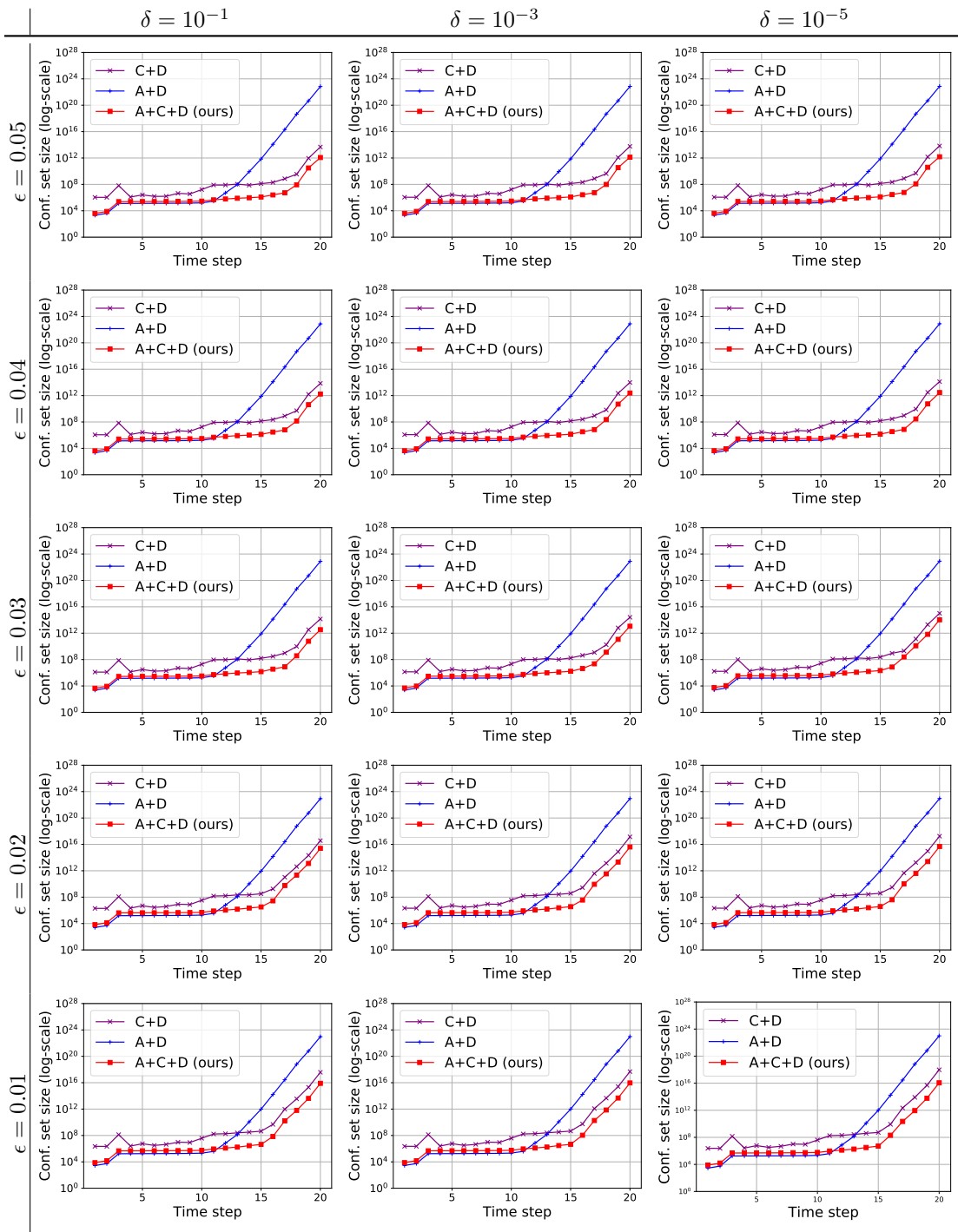

Table 8: Confidence set sizes for a neural network dynamics model trained on the half-cheetah environment, for varying $\epsilon, \delta$ and for $n = 5000$. The plots are as in Figure 3 (b).

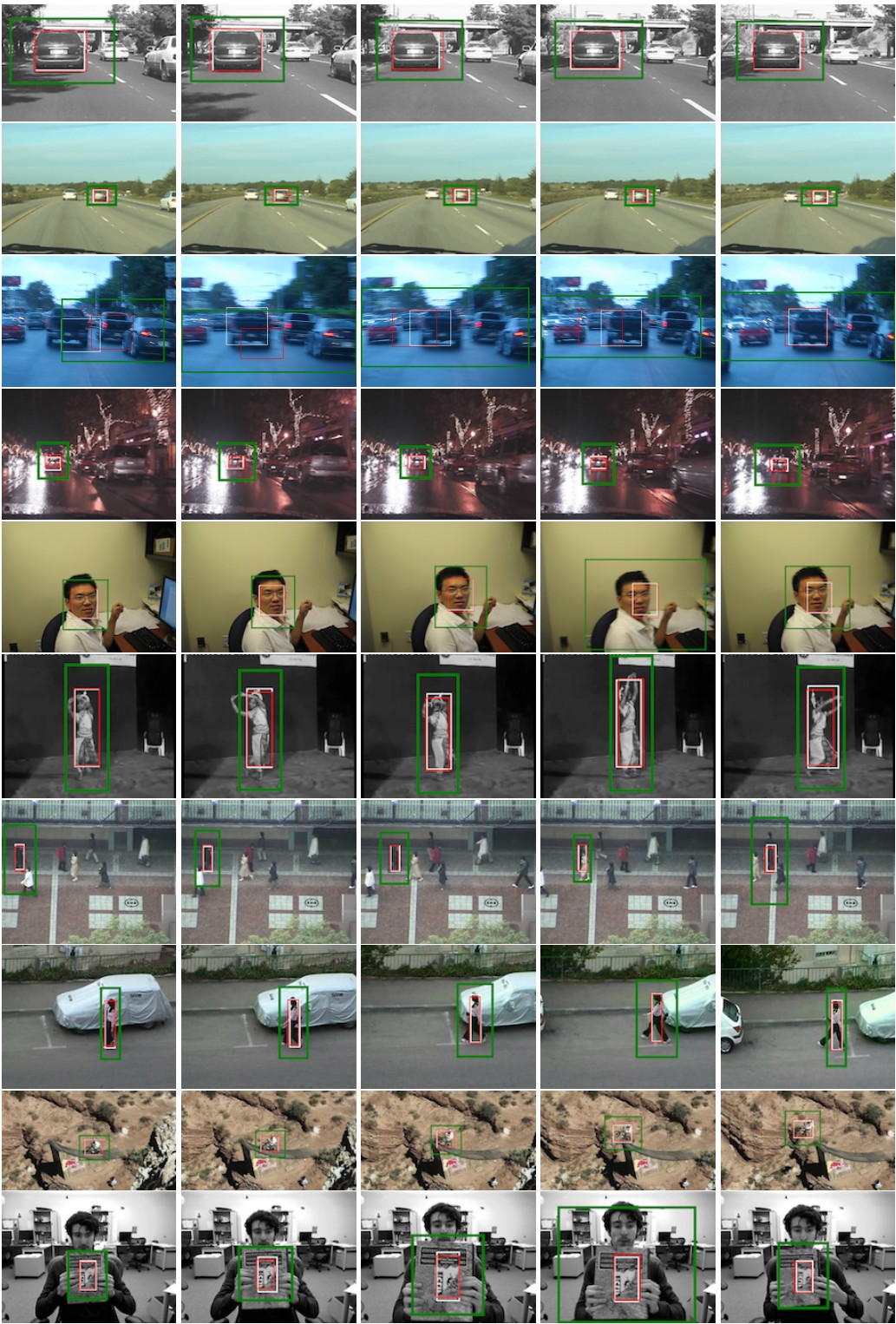

Table 9: Visualization of confidence sets for the tracking dataset (Wu et al., 2013), including the ground truth bounding box (white), the bounding box predicted by the original neural network (Held et al., 2016) (red), and the bounding box produced using our confidence set predictor (green). We have overapproximated the predicted ellipsoid confidence set with a box. Our bounding box contains the ground truth bounding box with high probability.

