# OpenReview forum: "PAC Confidence Sets for Deep Neural Networks via Calibrated Prediction"
_ICLR.cc/2020/Conference — Accept (Poster)_

### Official Review · AnonReviewer2 · 2019-10-22
**Official Blind Review #2**

**Rating:** 6

**Review:**

Summary: This paper presents an approach for generating confidence set predictions from deep networks. That is, the smallest set of predictions where the true answer is included in that set. Theory is used to derive an algorithm with PAC-style bounds on the population risk.


Overall Assessment: I like the core idea of this paper---developing region-based prediction algorithms with theoretical backing by taking advantage of a small calibration dataset and simple models of uncertainty. However, there are significant clarity and evaluation issues that must be addressed before the paper is ready for publication.  In current form, I rate the paper between Weak reject and strong reject overall.

Strengths:
+ Nice general direction.
+ New algorithm for confidence set prediction with theoretical guarantees.
+ Experiments show favourable performance vs a few ablations.

Weaknesses & Questions:
1. It is unclear to me why the temperature scaling component is needed. It does not actually change the ordering of the probabilities assigned to each class, so from what I can tell all it does is change the optimal T, but not the final performance.
2. The paper presents a bound on the population risk, but the experiments do not include a comparison of the expected worst-case error rates with the empirical error rates achieved on the test set. This should be corroborated.
3. The  description for the Model-based reinforcement learning subsection is impossible to follow due to ambiguity and lack of detail overall.
3.1. Some specific confusions about this section: Paper overloads f to be both a deterministic transition function, and also a distribution over possible states. It also switches between using x_{t+1} / x_t and x^\prime / x to mean the same thing. The notation used to describe the multi-step setting also seems to use "t" for two different things: the current time-step, and some arbitrary time-step in the past.
3.2. It is unclear what metric is being used to evaluate the state transition models---L2 distance between what? Given that the predictions are not point estimates, I would expect something that takes uncertainties into account.
5. For both sets of experiments (classification and RL), there are no alternative methods used as points of reference. There are a multitude of other approaches incorporating uncertainty into predictions, as mentioned in Section 1. A trivial baseline is to heuristically generates confidence sets by trusting the probabilities produced by the model and building a set out of the classes corresponding the top 1-\epsilon probability mass should be essential. This could be improved by applying difference calibration approaches to make the probabilities more trustworthy. Model ensembles provide another easy baseline. Such simple baselines are a minimum expectation, before even getting to state of the art alternatives.
6. There seem to be no vanilla regression experiment, only the harder-to-interpret RL experiment. EG: Since we already have a vision context: If you are doing facial age estimation , or interest point tracking, could one make a PAC prediction about the true age region/interest point location?
7. Overall the paper introduction misses some explanation on the motivating scenarios where such confidence set predictions are useful. One can perhaps imagine this for vision, but some help connecting the dots to how it could be useful in regression and RL would help.
8. It is claimed that Theorem 1 provides a "better" bound than the one based on the VC dimension. What is meant by better?

Minor comments:
* There are a couple of small mistakes in the proof of theorem 1. The \tilde{X} and \tilde{Y} in the definition of \tilde{Z}_{val} are in the wrong place. The "sum of k i.i.d. random variables" should be "sum of n i.i.d. random variables".
* In general, the proof is quite hard to follow. At times it was quite unclear how you get from one step to the next, because it relies on something shown several steps early, which is not referenced.
* Notation, in general, is a bit of an issue in this paper. See comments above, but also: switching between \theta and T for the parameter used in the confidence set predictor, some confusion between T and \hat{T}.
* It is unclear how the neural network used in model-based RL predicts a PSD covariance matrix.
* \hat{T} is not defined when it is first referenced (Section 3.3).
* Algorithm 1 appears several pages before it is referenced.

-------------- POST REBUTTAL ------------
I modify my score to 6: Weak accept.

**Experience Assessment:**

I have read many papers in this area.

**Review Assessment: Checking Correctness Of Derivations And Theory:**

I assessed the sensibility of the derivations and theory.

**Review Assessment: Checking Correctness Of Experiments:**

I assessed the sensibility of the experiments.

**Review Assessment: Thoroughness In Paper Reading:**

I read the paper at least twice and used my best judgement in assessing the paper.

---

> ### Author Response · Authors · 2019-11-14
> **Response (1/3)**
>
> First of all, we appreciate reviewer's valuable and constructive comments. In the following, we answer the reviewer's comments. The remaining answers and updated paper will be appeared.
>
>
> 1. It is unclear to me why the temperature scaling component is needed. It does not actually change the ordering of the probabilities assigned to each class, so from what I can tell all it does is change the optimal $T$, but not the final performance.
>
> --- The temperature scaling does not change the ordering of label probabilities for a single input $x$. However, the order of confidences of labels for different inputs can change. For example, consider two inputs $x$ and $x’$, for a problem with 10 labels. Assume that the label probabilities are
> $$
> f(x) = [0.1, 0.1, 0.1, ..., 0.1],
> f(x’) = [0.01, 0.99, 0.0, ..., 0.0]
> $$
> Now, if we take temperature $\tau$ very large (e.g., $\tau$ -> $\infty$), then the labels become roughly
> $$
> f_\tau(x) = [0.1, 0.1, 0.1, ..., 0.1],
> f_\tau(x’) = [0.49, 0.51, 0.0, ..., 0.0]
> $$
> Now, there are confidence sets that are achievable using f but not using $f_\tau$, and vice versa. For example, the confidence sets
> $$
> C(x) = \{0, 1, 2, ..., 9\},
> C(x’) = \{1\}
> $$
> can be achieved using $f$ (with $e^{-T} = 0.05$) but not using $f_\tau$. Conversely, the confidence sets
> $$
> C(x) = \{\},
> C(x’) = \{0, 1\}
> $$
> can be achieved using $f_\tau$ (with $e^{-T} = 0.25$) but not using $f$. Intuitively, we believe calibrating $\tau$ improves the ordering of probabilities across different inputs. Our experiments support this intuition: as we show, calibration helps reduce the confidence set size. We will clarify our paper to explain why calibrated prediction can improve the constructed confidence sets.
>
>
> 2. The paper presents a bound on the population risk, but the experiments do not include a comparison of the expected worst-case error rates with the empirical error rates achieved on the test set. This should be corroborated.
>
> --- We have added a comparison of the error rates achieved by our confidence sets on a held out test set. In particular, Figure 3 shows that our error rate is below the desired error rate $\epsilon$ on ImageNet on a held-out test set. The gap is likely due to the fact that the generalization bound is not tight. Nevertheless, the empirical error rate appears fairly close to the expected error rate. We also show results for HalfCheetah in Figure 4, which exhibit similar trends.
>
>
> 6. There seem to be no vanilla regression experiment, only the harder-to-interpret RL experiment. EG: Since we already have a vision context: If you are doing facial age estimation , or interest point tracking, could one make a PAC prediction about the true age region/interest point location?
>
> --- Thank you for the suggestions. We are happy to add a regression benchmark based on object tracking to the final version of the paper. Due to time limitations, for our current update, we instead applied our approach to two UCI regression datasets, namely, the mpg and student datasets. Results are shown in Figure 5. For the student dataset, we used parameters $n=100$, $\epsilon=0.1$, $\delta=0.05$; the constructed confidence sets achieve error of 0.0597 on a held-out test set. Similarly, for the MPG dataset, we used parameters $n=70$, $\epsilon=0.1$, $\delta=0.05$; the constructed confidence sets achieve error 0.125 on a held-out test set. We used larger choices of $\epsilon$ and $\delta$ since these datasets are very small, making it harder to get good bounds; also, in the MPG dataset, our bounds do not hold on a held-out test set (which is expected to happen with probability 0.05). These results demonstrate how our approach can be used to construct PAC confidence sets for regression problems.

---

> ### Author Response · Authors · 2019-11-14
> **Response (2/3)**
>
> (continue)
>
> 7. Overall the paper introduction misses some explanation on the motivating scenarios where such confidence set predictions are useful. One can perhaps imagine this for vision, but some help connecting the dots to how it could be useful in regression and RL would help.
>
> --- There are a number of reasons why confidence are useful. First, they are a measure of uncertainty, and can be used to inform safety critical decision making. For example, consider a doctor who uses prediction tools to help perform diagnosis. Having a confidence set would both help the doctor estimate the confidence of the prediction (i.e., smaller confidence sets imply higher confidence), but also give a sense of a large set of possible diagnoses.
>
> Alternatively, having a confidence set can be useful for reasoning about sets of possible outcomes. For instance, robots may use a confidence set over predicted trajectories to determine whether it is safe to act with high probability. As a concrete example, consider a self-driving car that uses a DNN to predict the path that a pedestrian might take. We would most likely expect the self-driving car to avoid the pedestrian with high probability. It can do so by avoiding all possible paths in the predicted confidence set.
>
> We will add a discussion of potential applications of confidence sets.
>
>
> 8. It is claimed that Theorem 1 provides a "better" bound than the one based on the VC dimension. What is meant by better?
>
> --- (This answer is updated in Response (3/3)) We use the terminology “better bound” in an intuitive sense. In particular, what we mean is that we expect the bounds will produce a smaller value of n (i.e., the number of samples) needed to achieve a given $\epsilon$ and $\delta$. Equivalently, for fixed $n$, $\epsilon$, and $\delta$, we expect to obtain smaller confidence sets.
>
> We intuitively expect that our bounds are tighter since we directly bound the error for our specific distribution rather than going through a general-purpose concentration inequality. For example, when $k = 0$, our direct bound is
> $$
> n_D(\epsilon, \delta) = O(\log\delta / \epsilon)
> $$
> whereas for the VC dimension bound, we have
> $$
> n_{VC}(\epsilon, \delta) = O(\log\delta / \epsilon^2)
> $$
> which is much larger. Using $k > 0$ can improve the size of confidence sets by handling outliers, at the expensive of increasing $n_D(\epsilon, \delta)$.
>
> Our experiments validate our intuition: the fact that we get smaller confidence sets in our experiments shows that at least for these problems, our direct bound is tighter than the VC dimension bound. Furthermore, we will add an empirical comparison of our direct bound and the VC dimension bound.

---

> ### Author Response · Authors · 2019-11-15
> **Response (3/3)**
>
> (continue)
>
> 3. The  description for the Model-based reinforcement learning subsection is impossible to follow due to ambiguity and lack of detail overall.
> 3.1. Some specific confusions about this section: Paper overloads f to be both a deterministic transition function, and also a distribution over possible states. It also switches between using x_{t+1} / x_t and x^\prime / x to mean the same thing. The notation used to describe the multi-step setting also seems to use "t" for two different things: the current time-step, and some arbitrary time-step in the past.
> 3.2. It is unclear what metric is being used to evaluate the state transition models---L2 distance between what? Given that the predictions are not point estimates, I would expect something that takes uncertainties into account.
>
> --- We have revised our description of the model-based reinforcement learning to address the comments and concerns, and have done our best to make it more precise and understandable. A part remains in Section 3.6 of the body of our paper, but a large part of the description is now in Appendix C due to space limitations.
>
>
> 5. For both sets of experiments (classification and RL), there are no alternative methods used as points of reference. There are a multitude of other approaches incorporating uncertainty into predictions, as mentioned in Section 1. A trivial baseline is to heuristically generates confidence sets by trusting the probabilities produced by the model and building a set out of the classes corresponding the top 1-\epsilon probability mass should be essential. This could be improved by applying difference calibration approaches to make the probabilities more trustworthy. Model ensembles provide another easy baseline. Such simple baselines are a minimum expectation, before even getting to state of the art alternatives.
>
> --- Thank you for the suggested baselines. By providing theoretical guarantees on its outputs, our approach is fundamentally different than these existing heuristics. For example, in safety-critical applications, having a guarantee that heuristically works is not sufficient (e.g., a self-driving car that heuristically avoids pedestrians would not be considered safe).
>
> Nevertheless, we have added results for two baselines along the lines you suggested: (i) using a threshold based on the probabilities of the original model, and (ii) using a threshold based on the calibrated probabilities. Specifically, for classification, we consider the confidence sets $C(x)$ constructed by ranking the labels in order of probability, and then choosing the top k such that $\sum_{i=1}^k f(y_i | x) \ge 1 - \epsilon$.
>
> We then compute the empirical confidence set error $|\{x | y* \not\in C(x)\}| / |\{x\}|$ on a held-out test set -- i.e., the fraction of points for which the true label is not in the confidence set. As can be seen, the errors for the baselines do not meet the desired error rate $\epsilon$; for the baseline, its error is more than double the desired rate. In contrast, our confidence sets, while larger in size, have the desired error rate <= $\epsilon$ with probability at least $1 - 10^{-5}$.
>
> We have included the detailed description in Appendix D.1 for both classification and regression.
>
>
> 8. It is claimed that Theorem 1 provides a "better" bound than the one based on the VC dimension. What is meant by better?
>
> --- (updated answer) We use the terminology “better bound” in two senses. First, we expect to construct smaller confidence sets using our bound. We also mean it in the more standard sense that we can obtain the same ($\epsilon$, $\delta$) PAC guarantee using fewer samples (i.e., a smaller value of $n$).
>
> Our experiments validate our intuition: the fact that we get smaller confidence sets in our experiments shows that at least for these problems, our direct bound is tighter than the VC dimension bound. We can also give some more analytical insights. We have added a discussion to Appendix A.2 in our paper.
>
>
> =========== response for the minor comments
> Thanks for these detailed comments. We were trying to resolve the clarification issues, as the reviewer pointed out. For the question on "how neural network used in model-based RL predicts a PSD covariance matrix", we use a neural network that predicts a covariance matrix of the form $diag(\sigma_1^2, ..., \sigma_n^2)$ -- i.e., an axis-aligned Gaussian, as proposed in (Chua et al., 2018).

---

### Official Review · AnonReviewer1 · 2019-10-23
**Official Blind Review #1**

**Rating:** 6

**Review:**

The paper proposes an algorithm combining calibrated prediction and generalization to construct confidence sets for deep neural networks with PAC guarantees.
The main novelty is that existing approaches do not come with PAC guarantees
Following (Platt et al., 1999) and (Guo et al., 2017), the calibration of the learned model is controlled by its temperature. In particular, the proposed approach exploits another (small) training dataset to learn a temperature that gives the best calibration.
An efficient algorithm for constructing confidence sets that are "small in size" is also proposed.
The framework is introduced for the classification, regression and reinforcement learning tasks.

I vote for acceptance for the following reasons:
- The paper is well-written, the motivation and comparison to related work is also clear.
- The paper is very solid theoretically and experimentally. A theoretical analysis is provided, and practical implementations are proposed to deal with scalability issues. VC generalization bounds are studied in detail.

Concerning experiments that are not about Reinforcement learning, the paper proposes different strategies to learn a ResNet architecture for ImageNet. One might argue that the different results might be only valid for the chosen architecture and dataset.
Although the study for that architecture and (large scale) dataset is exhaustive, it might be interesting to study if the reported results are also valid on (smaller?) datasets and other architectures.

**Experience Assessment:**

I do not know much about this area.

**Review Assessment: Checking Correctness Of Derivations And Theory:**

I assessed the sensibility of the derivations and theory.

**Review Assessment: Checking Correctness Of Experiments:**

I assessed the sensibility of the experiments.

**Review Assessment: Thoroughness In Paper Reading:**

I read the paper at least twice and used my best judgement in assessing the paper.

---

> ### Author Response · Authors · 2019-11-13
> **Response**
>
> Thank you for the suggestions regarding additional experiments. We are happy to add additional experiments on smaller datasets and different architectures (e.g., AlexNet and VGG-19) with the ImageNet dataset. For the current revision, we have added results for some very small UCI datasets. We will subsequently add results on other small datasets, including MNIST.

---

### Official Review · AnonReviewer3 · 2019-10-25
**Official Blind Review #3**

**Rating:** 6

**Review:**

This paper propose to estimate the confidence sets for deep neural networks with PAC guarantees by combining calibrated prediction and generalization bounds from learning theory. Reliable confidence set estimation can be very useful for applications in safe-critical domains, and a paper advancing knowledge in this area is certainly welcome. The paper is clearly presented and well motivated. In addition to give a direct generalization bound that trades off between realizable setting and the VC bound, it also proposes an algorithm for predict the confidence set in practice. Experimental results in image classification and reinforcement learning illustrate the effectiveness of the proposed algorithm. One thing that may need further clarify is that the derived bound and algorithm is general in that it can be applied to any estimator f_\theta using 4 and 5. What kind of structural characteristic of neutral nets that are explicitly exploited in the algorithm in particular? Will this consideration further tighten the bound? Small typo: under Equation 5, Z^\prime_{train} is oftentimes the Z_{val}.

**Experience Assessment:**

I have read many papers in this area.

**Review Assessment: Checking Correctness Of Derivations And Theory:**

I assessed the sensibility of the derivations and theory.

**Review Assessment: Checking Correctness Of Experiments:**

I assessed the sensibility of the experiments.

**Review Assessment: Thoroughness In Paper Reading:**

I read the paper at least twice and used my best judgement in assessing the paper.

---

> ### Author Response · Authors · 2019-11-13
> **Response**
>
> Thank you for your comments and suggestions. The only property exploited by our algorithm is that the neural network outputs a score representing its confidence in each label. We will emphasize that the derived bound and the algorithm in the paper work with any estimator f_\theta that satisfies this condition. While it may be possible to exploit additional properties of the estimator to tighten these bounds, it has traditionally been very hard to do so due to the challenges in providing any kind of theoretical bounds on generalization of deep neural networks. In contrast, we believe approaches based on treating the neural network as a blackbox are much more practical; in addition, they have the advantage of being more general and more scalable. Finally, we will fix the typos you mentioned; thanks for pointing them out.

---

### Author Response · Authors · 2019-11-15
**Paper Revision**

Dear Reviewers,

We thank the reviewers for their helpful comments and suggestions. We have made the following changes/additions to our paper:
-- We have added a paragraph to the introduction motivating the need for PAC confidence sets (Section 1)
-- We have improved the exposition on model-based RL (Section 3.6, Appendix C)
-- We have added a section on why calibration can improve our confidence sets (Appendix A.1)
-- We have added a section describing the sense in which our direct bound is better than the VC dimension bound (Appendix A.2)
-- We have improved our exposition of the proof (Appendix B)
-- We have added results comparing to baselines that do not provide theoretical guarantees (Appendix D.1)
-- We have added results on some small regression benchmarks (Appendix D.2)
-- We have made some improvements to the baselines in our RL benchmark, and have updated the results accordingly.

---

### Decision · Program_Chairs · 2019-12-19

**Decision:**

Accept (Poster)

**Comment:**

This paper describes a method for bounding the confidence around predictions made by deep networks. Reviewers agree that this result is of technical interest to the community, and with the added reorganization and revisions described by the authors, they and the AC agree the paper should be accepted.